# Learned adaptive multiphoton illumination microscopy for large-scale immune response imaging

Henry Pinkard [1,2,3,4✉], Hratch Baghdassarian [5], Adriana Mujal [5], Ed Roberts [5], Kenneth H. Hu[5], Daniel Haim Friedman[6], Ivana Malenica [3,7], Taylor Shagam[5], Adam Fries[5], Kaitlin Corbin[5], Matthew F. Krummel [5,8] & Laura Waller [2,3,8]

Multiphoton microscopy is a powerful technique for deep in vivo imaging in scattering samples. However, it requires precise, sample-dependent increases in excitation power with depth in order to generate contrast in scattering tissue, while minimizing photobleaching and phototoxicity. We show here how adaptive imaging can optimize illumination power at each point in a 3D volume as a function of the sample's shape, without the need for specialized fluorescent labeling. Our method relies on training a physics-based machine learning model using cells with identical fluorescent labels imaged in situ. We use this technique for in vivo imaging of immune responses in mouse lymph nodes following vaccination. We achieve visualization of physiologically realistic numbers of antigen-specific T cells (~2 orders of magnitude lower than previous studies), and demonstrate changes in the global organization and motility of dendritic cell networks during the early stages of the immune response. We provide a step-by-step tutorial for implementing this technique using exclusively open-source hardware and software.

[1] Department of Electrical Engineering and Computer Sciences, University of California, Berkeley, CA, USA. [2] Computational Biology Graduate Group, University of California, Berkeley, CA, USA. [3] Berkeley Institute for Data Science, Berkeley, CA, USA. [4] University of California San Francisco Bakar Computational Health Sciences Institute, San Francisco, CA, USA. [5] Department of Pathology, University of California, San Francisco, San Francisco, CA, USA. [6] Department of Bioengineering, University of California, Berkeley, CA, USA. [7] Division of Biostatistics, University of California, Berkeley, CA, USA. [8]These authors contributed equally: Matthew F. Krummel, Laura Waller. ✉email: hbp@berkeley.edu

maging of cells in vivo is an essential tool for understanding the spatiotemporal dynamics that drive biological processes. For highly scattering tissues, multiphoton microscopy (MPM) is unique in its ability to image deep into intact samples (200 μm–2 mm, depending on the tissue). Because of the non-linear relationship between excitation light power and fluorescence emission, scattered excitation light contributes negligibly to the detected fluorescence emission. Thus, localized fluorescent points can be imaged deep in a sample in spite of a large fraction of the excitation light scattering away from the focal point, by simply increasing the incident excitation power[1] (Fig. 1a).

The dual problems photobleaching and photodamage are an inescapable part of every fluorescence imaging experiment. The concept of a "photon budget" is often used to express the inherent trade-offs between sample health, signal, spatial resolution, and temporal resolution, and a widely pursued goal is to make microscopes that are as gentle as possible on sample while still generating the contrast necessary for biological discovery[2]. These problems are an especially acute concern in MPM since, unlike in single-photon fluorescence, they increase supra-linearly with respect to the intensity of fluorescence emission[3,4].

When imaging deep into a sample using MPM, excitation light focusing to different points in the sample will be subjected to

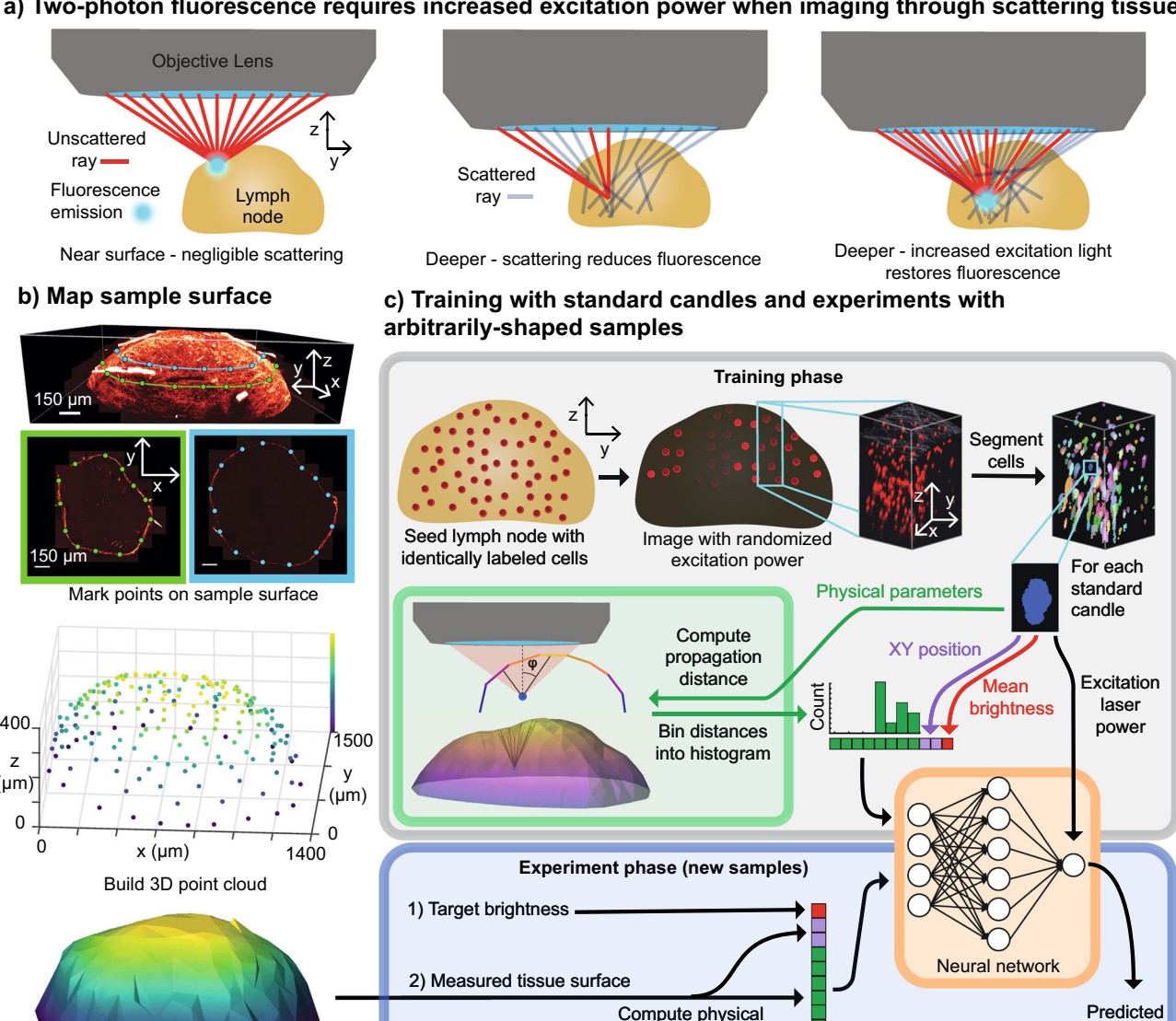

**Fig. 1 Learned adaptive multiphoton illumination (LAMI). a** In vivo multiphoton microscopy requires increasing laser power with depth to compensate for the loss of fluorescence caused by excitation light being scattered. **b** Our LAMI method uses the 3D sample surface as input to its neural network. We map it by selecting points on XY image slices at different Z positions (top) to build up a 3D distribution of surface points (middle) that can be interpolated. **c** Training uses samples seeded with cells with the same fluorescent label (standard candles), which is imaged with a random amount of power. A 3D segmentation algorithm then isolates the voxels corresponding to each standard candle. The mean brightness of these voxels, position in XY field of view, and a set of physical parameters (a histogram of propagation distances through the tissue to the focal point at a specific angle of inclination to the optical axis ($\phi$)) are concatenated into a single vector for each standard candle. The full set of these vectors is used to train a neural network that predicts excitation laser power. (Bottom) After training, subsequent samples need not be seeded with standard candles. The network automatically predicts point-wise excitation power as a function of the sample geometry and a user-specified target brightness.

different amounts of scattering, and the excitation laser power must be increased in order to maintain signal. Failing to increase sufficiently will lead to the loss of detectable fluorescence. Increasing too much subjects the sample to unnecessary photobleaching and photodamage, with the potential to disrupt or alter the biological processes under investigation. If done improperly, this can even result in visible burning or destruction of the sample (Supplementary Movie S1). This problem is especially pronounced in highly scattering tissue (e.g., in lymph nodes) because the appropriate excitation power has more rapid spatial variation compared to less scattering tissues.

Adaptive optics (AO) represents one strategy for addressing this challenge[5,6]. By pre-compensating the shape of the incident excitation light wavefront based on the scattering properties of the tissue, the fraction of incident light that reaches the focal point increases, lessening the need to increase power with depth. However, AO still suffers from an exponential decay of fluorescence intensity with imaging depth when using constant excitation[1,6], so an increase in incident power with depth is still necessary.

Alternatively, instead of minimizing scattering with AO, adaptive illumination techniques modulate excitation light intensity to ensure the correct amount reaches the focus. To make the best use of a sample's photon budget, these methods should increase power to the minimal level needed to yield sufficient contrast, but no further than this to avoid the effects of photobleaching and photodamage.

Most commercial and custom-built multiphoton microscopes have some capability to increase laser power with depth, either using an exponential profile or an arbitrary function. For a flat sample (e.g., imaging into brain tissue through a cranial window), these techniques work well. The profile of fluorescence decay with depth can be approximated by an exponential or heuristically defined for an arbitrary function, by focusing to different depths in a sample and manually specifying increases. However, this task is more complex for a curved or irregularly-shaped sample, in which such profiles shift as the height of the sample varies and change shape in different areas of the sample.

A more advanced class of methods for adapting illumination uses feedback from the sample during imaging. This strategy has been employed previously in both confocal[7] and multiphoton[8,9] microscopy. The basic principle is to implement a feedback circuit between the microscope's detector and excitation modulation, such that excitation light power is turned off at each pixel once a sufficient number of photons have been detected. However, this approach does not account for fluorophore brightness and labeling density; thus, it is impossible to disambiguate weak fluorophores (e.g., a weakly expressed fluorescent protein) receiving a high dose of incident power from strong fluorophores (e.g., a highly expressed fluorescent protein) receiving a low dose. Not only does this run the risk of unnecessarily depleting the photon budget, it can also lead to over-illumination and photodamage if left unchecked. To prevent photodamage, a heuristic user-specified upper bound is set to cap the maximum power. Such an upper bound can vary by over an order-of-magnitude when imaging into highly scattering thick samples. Thus, applying this approach to image 100s of μms deep in such samples still requires additional prior knowledge about the attenuation of fluorescence in different parts of a sample.

The difficulties of adaptive illumination in non-flat samples thus create several problems. First, the range over which sufficient contrast can be generated is limited to the sub-region where an appropriate function to modulate power can be ascertained and applied by the hardware. Second, incorrect modulations can deplete the photon budget and cause unnecessary photodamage, with unknown effects on the processes under observation.

In intravital imaging of the popliteal lymph node, an important model system for studying vaccine responses, the constraint on imaging volumes imparts an unfortunate bias. Previous studies of T cell dynamics in intact lymph nodes have increased the density of transferred monoclonal T cells in order to achieve sufficient numbers for visualization ($10^6$ or more) within the limited imaging volume of MPM. This number is 2–3 orders-of-magnitude more than the number of reported clonal precursor T cells ($10^3-10^4$) under physiological conditions[10,11]. It is well established that altering precursor frequencies changes the kinetics and outcome of the immune responses[12–15], but it is unknown how these alterations might have affected the conclusions of previous studies.

Here, we describe a data-driven technique for learning the appropriate excitation power as a function of the sample shape, and provide a simple hardware modification to an multiphoton microscope that enables its application. Our method can provide 10–100× increase in the volume to which appropriate illumination power can be applied in curved samples such as lymph nodes, and a reproducible way to automatically apply the minimal illumination needed to observe structures of interest, thereby conserving the photon budget and minimizing the perturbation to the sample induced by the imaging process. Significantly, our method neither requires the use of additional fluorescence photons to perform calibration on each sample, nor specialized sample preparation to introduce fiducial markers.

The method uses a one-time calibration experiment to learn the parameters of a physics-based machine learning model that captures the relationship between fluorescence intensity and incident excitation power in a standardized sample, given the sample's shape. On subsequent experiments, this enables continuous adaptive modulation of incident excitation light power as a focal spot is scanned through each point in the sample. We describe a simple hardware modification to an existing multiphoton microscope that enables modulation of laser power as the excitation light is scanned throughout the sample. This modification costs <$50 for systems that already have an electro-optical or accousto-optic modulator, as most modern multiphoton systems do. We call our technique learned adaptive multiphoton illumination (LAMI).

Our central insight is inspired by the idea of "standard candles" in astronomy[16], where the fact that an object's brightness is known a priori allows its distance to Earth to be inferred based on its apparent brightness. Analogously, we hypothesize that by measuring the fluorescence emission of identically labeled cells ("standard candles") at different points in a sample volume under different illumination conditions, we could use a physics-based neural network to learn an appropriate adaptive illumination function that could predicted from sample shape alone.

Applying LAMI to intravital imaging of the mouse lymph node, we first show that the learned function generalizes across differently shaped samples of the same tissue type (e.g., one mouse lymph node to another). Moving to a new tissue type, which would attenuate light differently, would require a new calibration experiment. After a one-time calibration experiment, the trained neural network can be used to automatically modulate excitation power to the appropriate level at each point in new samples, enabling dynamic imaging of the immune system with single-cell resolution across volumes of tissue more than an order-of-magnitude larger than previously described. Unlike previous studies that artificially increased the number of monoclonal precursor T cells to >$10^6$ (2 orders-of-magnitude greater than typical physiological conditions) in order to visualize them in a small imaging volume[17,18], we image physiologically realistic ($5 \times 10^4$ transferred) cell frequencies.

## Results

**Learning illumination power as a function of shape**. The detected fluorescence intensity at a given point results from a combination of two factors: (1) the sample-dependent physics of light propagation (e.g., scattering potential of the tissue, fraction of emitted photons that are detected, etc.), which are difficult to model a priori due to heterogeneity in sample shapes. (2) The fluorescent labeling (e.g., the type and local concentration of fluorophores), a nuisance factor that makes it difficult to disambiguate weak fluorophores receiving a high dose of incident power from strong fluorophores receiving a low dose.

Our method relies on the fact that, if fluorescence labeling of distinct parts of the sample is, on average, constant (i.e., "standard candles"), we can separate out the effects of fluorescence strength and tissue-dependent physics by performing a one-time calibration to learn the effect of only the tissue-dependent physics for a given tissue type. The calibrated model captures the effects of the physics relating excitation power, detected fluorescence, local sample curvature, and position in the XY field of view (FoV), which includes optical vignetting effects. By generating a dataset consisting of points with random distributions over these variables, we can learn the parameters of a statistical model to predict excitation power as a function of detected fluorescence, sample shape, and position. On subsequent experiments in different samples of the same type, the model can predict the excitation power required to achieve a desired level of detected fluorescence for each point in the sample based only on sample shape and XY position.

The standard candle fluorophores are only necessary during the calibration step. In the mouse lymph node, we introduce them by transferring genetically identical, identically labeled (with either cytosolic fluorescent protein or dye) lymphocytes, which then migrate into lymph nodes and position themselves throughout its volume. Although there are certainly stochastic differences in labeling density between individual cells (e.g., noise in expression of fluorescent proteins), the neural network estimates the population mean, so as long as these differences are not correlated with the cells' spatial locations, they will not bias the calibration.

An important consideration is what type of statistical model will be used to predict excitation power. One possibility is a purely physics-based model. We developed such a model using principles of ray optics by computing the length each ray travels through the sample and its probability of scattering before reaching the focal point (Supplementary Fig. S4). When one must predict excitation in real time, however, this model is too computationally intensive (~1 s per focal point). To circumvent the problem, the model parameters can be pre-computed, but this requires the assumption of an unrealistic, simplified sample shape, thus introducing a sample-dependent source of model mismatch. On top of this, there may be additional sources of model mismatch, such as a failure to account for wave-optical effects, inhomogeneous illumination across the FoV, spatial variation in attenuation of fluorescence signal, etc.

Given this model mismatch, we found that a physics-based neural network was a better solution. Unlike the purely physics-based model, a physics-based neural network is a flexible function approximator that can be easily adapted to incorporate additional relevant physical quantities into its predictions. For example, accounting for variations in brightness across a single FoV would require building optical vignetting effects into a physical model, whereas a neural network can simply take position in the FoV as an input and learn to compensate for these effects. Importantly, a small neural network can make predictions quickly (~1 ms per focal point) and is thus suitable for real-time application.

The neural network makes its predictions based on measurements of the sample shape that capture important parameters of the physics of fluorescence attenuation. To measure these parameters, points were hand selected on the sample surface in XY images of a focal stack to generate a set of 3D points representing the outer shape of the sample (Fig. 1b). These points were interpolated in 3D in a piece-wise linear fashion to create a 3D mesh of the sample surface. In MPM, the distance light traveling through tissue is an important quantity, as both the fraction of excitation light that attenuates from scattering/absorption and the fraction of fluorescence emission that absorbs are proportional to the negative exponential of this distance[1], assuming homogeneous scattering. We thus reasoned that measuring the full distribution of path lengths (i.e., every ray within the objective's numerical aperture—the same starting point of the ray optics model) would provide an informative parameterization to predict fluorescence attenuation. Empirically, we found that the full distribution of distances was not needed to achieve optimal predictive performance (based on error on a held out set of validation data during neural network training), and that measuring 12 distances along lines with a single angle of inclination relative to the optical axis was sufficient (Fig. 1c, green box). We encode the assumption that the optical system is rotationally symmetric about the optical axis by binning the measured distances into a histogram. The counts of this histogram were used in the feature vector fed into the neural network.

The neural network takes inputs of mean standard candle brightness, local sample shape, and position within the XY FoV and outputs a predicted excitation power (Fig. 1c, orange box). The network is trained using a dataset with a single standard candle cell that was imaged with a random, known amount of excitation power. Neural networks are excellent interpolators and poor extrapolators, so we ensured that the random excitation power used in training induced a range of brightness spanning too-dim-to-see to unnecessarily bright (Fig. 1c, top middle and Supplementary Movie S2). Unlike contemporary deep neural networks[19], the prediction only requires a very small network with a single hidden layer (a $10^4$–$10^6$ reduction in number of parameters compared to state-of-the-art deep networks). Once trained, the network can then be used with new samples to predict the point-wise excitation power needed for a given level of brightness (Fig. 1c, bottom). In a shot noise-limited regime, the signal-to-noise ratio (SNR) is proportional to $\sqrt{N}$, where $N$ is the number of photons collected, while brightness is proportional to $N$ (assuming a detector with a linear response). Thus, this brightness level can be interpreted at $SNR^2$ for a constant level of labeling density. After the one-time network training with standard candles, experiments can be fluorescently labeled without standard candles, and only the sample shape is needed to predict excitation power.

**Modulating excitation light across field of view**. The appropriate excitation power often varied substantially across a single $220 \times 220$ μm FoV—visibly so when imaging curved edges of the lymph node where the sample was highly inclined relative to the optical axis, thereby including both superficial and deep areas of the lymph node (Supplementary Fig. S1). The trained network predicted very different excitation powers from one corner of the FoV to another in such cases. In order to be able to deliver the correct amount of power, we need to be able to spatially pattern excitation light at different points within a single FoV as the microscope scans through all points in 3D. To accomplish this, we designed a time-realized spatial light modulator (TR-SLM) capable of modulating excitation laser power over time as it raster

scans a single FoV (Supplementary Figs. S1–S3). Unlike a typical SLM, we leverage the point scanning nature of multiphoton microscopy to achieve 2D spatial patterning by changing the voltage of an electro-optic modulator (EOM) at a rate faster than the raster scan rate in order to spatially pattern the strength of excitation. 3D spatial patterning is achieved by applying different 2D patterns when focused to different depths. This method has the advantage or avoiding reflection or transmission losses associated with SLMs, thereby maintaining use of the full power of the excitation laser. The TR-SLM was built using an Arduino-like programmable micro-controller connected to a small op-amp circuit that output a voltage to an EOM, allowing it to retrofit an existing multiphoton microscope for less than $50.

**Generalization across samples**. To validate the performance of LAMI and demonstrate that it can generalize across samples, we trained the network on a single lymph node and tested on a new, differently shaped lymph node (Fig. 2a). The test lymph node was seeded with a variety of fluorescent labels and imaged ex vivo to

eliminate the possibility of motion artifacts associated with intravital microscopy. The surface of the test lymph node was mapped as described previously (Fig. 1b). Several different desired brightness levels were tested to find one with appropriate signal. For comparison, we imaged the test lymph node with a constant excitation power, with an excitation power predicted by a ray optics model, and with LAMI (Fig. 2b). Since a full ray optics model was too computationally intensive to be computed at each point in real time, we made the a priori assumption of a perfectly spherical sample for our ray optics model comparison. With constant excitation, fluorescence intensity rapidly decayed after the first 25–50 μm. The ray optics model, which modulated illumination based on both depth and curvature, provided visualization of a much larger area, but still exhibited visible heterogeneity, including areas with little to no detectable fluorescence. This makes sense given that the lymph node was not perfectly spherical, which the model had assumed. LAMI provided clear visualization of cells throughout the volume of the lymph node (Fig. 2b and Supplementary Movie S3), up to the depth limit imposed by the maximum power of the excitation

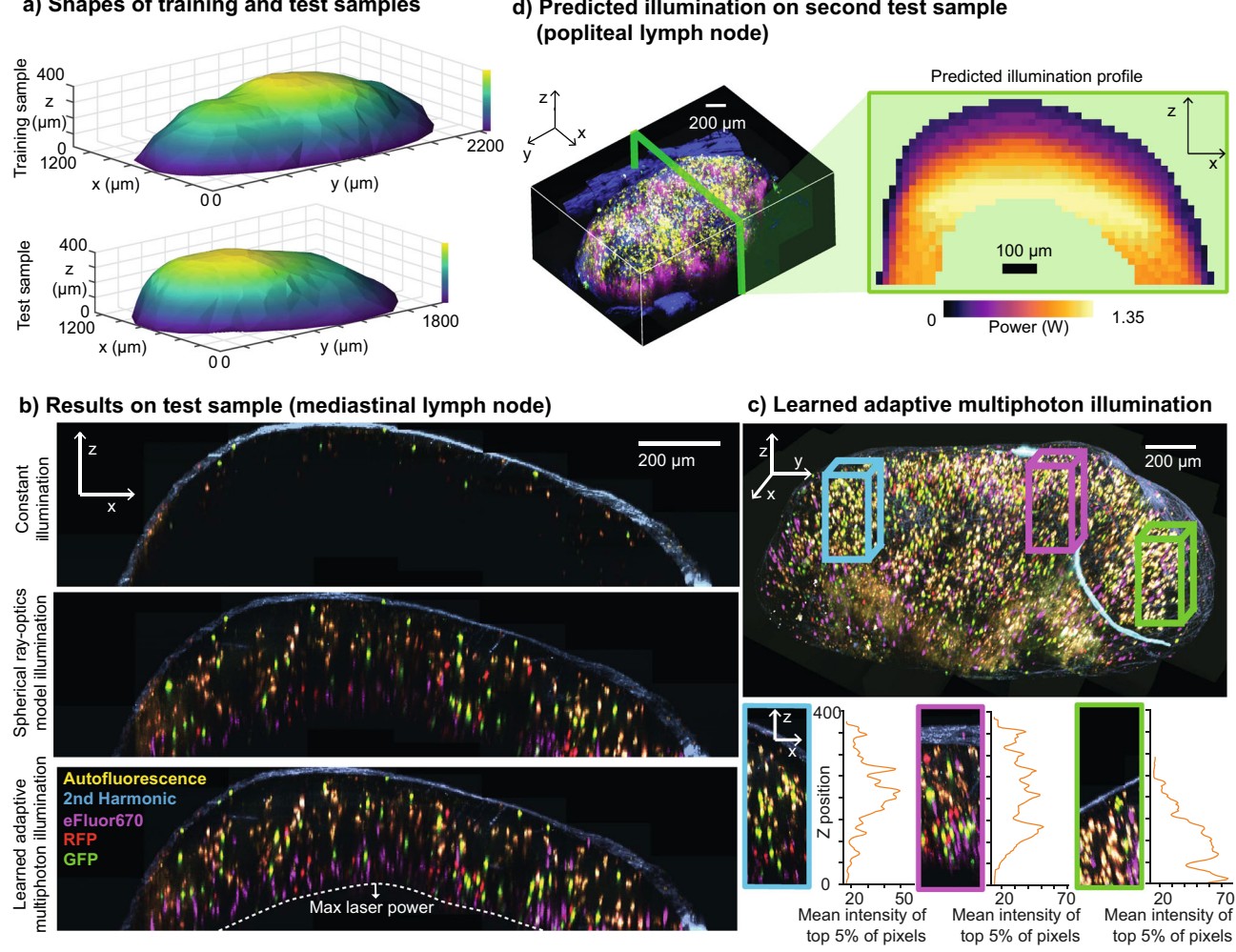

**Fig. 2 Validation of LAMI on lymph node samples. a** The surface shapes of lymph nodes used for (top) training with standard candles and (bottom) testing. **b** Results with constant illumination power, illumination power predicted by the ray optics model that assumed a perfectly spherical shape, and illumination power predicted by LAMI in the test sample, which had been seeded with lymphocytes labeled with GFP (green), RFP (red), and eFluor670 (magenta). Constant illumination rapidly attenuates the signal with depth. The ray optics model generates contrast throughout the volume, but has visible non-uniformity and areas where the signal from cells is entirely missing. In contrast, LAMI gives good signal throughout the imaging volume up to the maximum excitation laser power. **c** A 3D view of the LAMI-imaged lymph node, with several XZ projections of representative areas with different surface curvature. Plots show Z-position vs mean intensity of top 5% of pixels to demonstrate good signal is maintained with depth using LAMI. **d** Popliteal lymph node imaged with LAMI along with XZ cross-section of predicted illumination.

laser on our system of around 300–350 μm (Fig. 2e). In interpreting these data, it is important to note that the images were taken sequentially, so some movement of individual cells between images is expected. Similar performance was maintained even on lymph nodes with irregular, multi-lobed shapes (Supplementary Movie S4).

Unlike a flat sample, where fluorescence attenuates with depth following an exponential function[1], a curved, convex sample such as a lymph node has a sub-exponential decay with depth (Supplementary Fig. S5). To better understand how the appropriate excitation power changes across the sample, we visualized the predictions of the neural network across space (Fig. 2c). This prediction can be used to make a quantitative comparison of volumes of the sample to which appropriate excitation power can be delivered with LAMI vs other common adaptive excitation strategies in MPM (Supplementary Fig. S6).

In order to conduct LAMI experiments on in vivo samples, we added two additional data processing steps: (1) correcting motion artifacts, which are an inescapable feature of intravital imaging (Supplementary Figs. S7 and S8 and Supplementary Movies S5 and S6) and (2) developing a pipeline for identifying and tracking multiple cell populations across time (Supplementary Figs. S8 and S9). The latter used an active machine learning[20] framework to amplify manual data labeling, which led to a 40× increase in the efficiency of data labeling compared to labeling examples at random (Supplementary Fig. S8).

**In vivo lymph node imaging under physiological conditions.** Using our system, we conducted a biological investigation of a common model system for response to vaccination, in vivo imaging of a murine popliteal lymph node in an anesthetized mouse. Subunit vaccines are a clinically used subset of vaccines in which patients are injected with both a part of a pathogen (the antigen/subunit) and an immunogenic molecule to elicit a protective immune response (the adjuvant). A common model system for these consists of mice being immunized with Ovalbumin, a protein in egg whites, as a model antigen and lipopolysaccharide adjuvant. Before immunization, fluorescently labeled T cells that specifically respond to Ovalbumin (monoclonal OT-I and OT-II T cells) are also transferred to the host mouse so that their antigen-specific behavior can be observed in relation to antigen-presenting cells in the local lymph node where the initial immune response occurs.

Typically, these experiments can only image a small volume of the lymph node at once. In order to visualize a sufficient number of antigen-specific T cells, previous studies transferred 2–3 orders-of-magnitude more monoclonal cells than would typically exist under physiological conditions, a modification that is well established to alter the dynamics and outcomes of immune responses[12–15]. With our LAMI technique, we can deliver the correct excitation power to 10–100× larger volume of tissue (the exact number depends on what baseline, as described in Supplementary Fig. S6, Methods), so the perturbation of introducing a physiologically unrealistic number of cells is no longer needed. We use an endogenous population of fluorescently labeled antigen-presenting cells, type I conventional dendritic cells labeled with Venus under the XCR1 promoter[21].

Twenty-four hours after immunization with lipopolysaccharide, the type I conventional dendritic cell network exhibited a marked reorganization (Supplementary Fig. S10 and Supplementary Movies S7 and S8), with XCR1+ cells clustering closer to each other and moving from a more even distribution throughout various areas of the lymph node into primarily the paracortex. We found that these clusters of dendritic cells were located primarily around OT-I (CD8 T cells specific to Ovalbumin)

rather than OT-II (CD4 T cells specific to Ovalbumin) or polyclonal T cells, and closer to high endothelial venules than in the control condition.

Imaging and tracking dendritic cells in a control condition and at 24 h after immunization revealed that this reorganization was accompanied by a change in motility, with dendritic cells at the 24 h time point moving both more slowly and in a subdiffusive manner, thus confining themselves to smaller areas (Supplementary Movie S9) compared to the more exploratory behavior of the control condition (Fig. 3a). This decrease in average motility appeared global with respect to anatomical subregions and the local density of other dendritic cells (Supplementary Fig. S11). These changes in dendritic cell motility were also accompanied by changes in T cell motility in an antigen-specific manner. OT-I T cells, which appeared at the center of dense clusters of dendritic cells, showed the most confined motility compared to polyclonal controls, while OT-II cells were often found on the edges of these clusters with slightly higher motility (Fig. 3b and Supplementary Movie S7).

To understand how this reorganization takes place, we next imaged lymph nodes 5 h after immunization. Although dendritic cell motility has not yet changed at this time point, the increasing formation of clusters is detectable on the timescale of an hour (Fig. 3c). Over time, new clusters appeared to form both from spatially separated dendritic cells moving toward one another, and from isolated dendritic cells moving toward and joining larger existing clusters (Supplementary Movies S10 and S11).

These findings reveal that there is a marked difference in the location and behavior of dendritic cell networks encountered by T cells that enter an inflamed lymph node at the beginning vs the later stages of an immune response. Notably, they also show that the larger-scale dendritic cell reorganization precedes the T cell activation-induced motility arrest that we and others observe amongst antigen-specific T cells at the 24-h time point.

We speculate that this increased local concentration of dendritic cells may be necessary for rare, antigen-specific T cells to find one another and form the homotypic clusters required for robust immunological memory[22]. The reorganized environment could be an important factor in the difference in differentiation fate of T cells that enter lymph nodes early vs late in immune responses[23].

## Discussion
We have demonstrated how a computational imaging MPM approach, LAMI, provides a rigorous, data-driven approach for adapting illumination to achieve sufficient signal-to-noise without over-illuminating the sample. This removes an important source of human bias, heuristic adjustment of illumination, and thereby enables automated, reliable, and reproducible imaging experiments. Significantly, it neither requires specialized sample preparation nor additional calibration images that deplete the sample's photon budget. This technology enables imaging experiments with more physiological conditions. In this work, we demonstrate an example of lymph node imaging with 100× lower T cell frequencies.

LAMI is most useful for highly scattering samples with non-flat surfaces (e.g., lymph nodes, large organoids or embryos), which have complicated functions mapping shape to excitation. Applying LAMI to other tissues will require development of sample-specific standard candles. There are many possibilities for these—the only strict conditions are having a labeling density that is not location specific and that individual standard candles can be spatially resolved. Some possibilities for standard candles include genetically encoded cytoplasmic fluorophores or

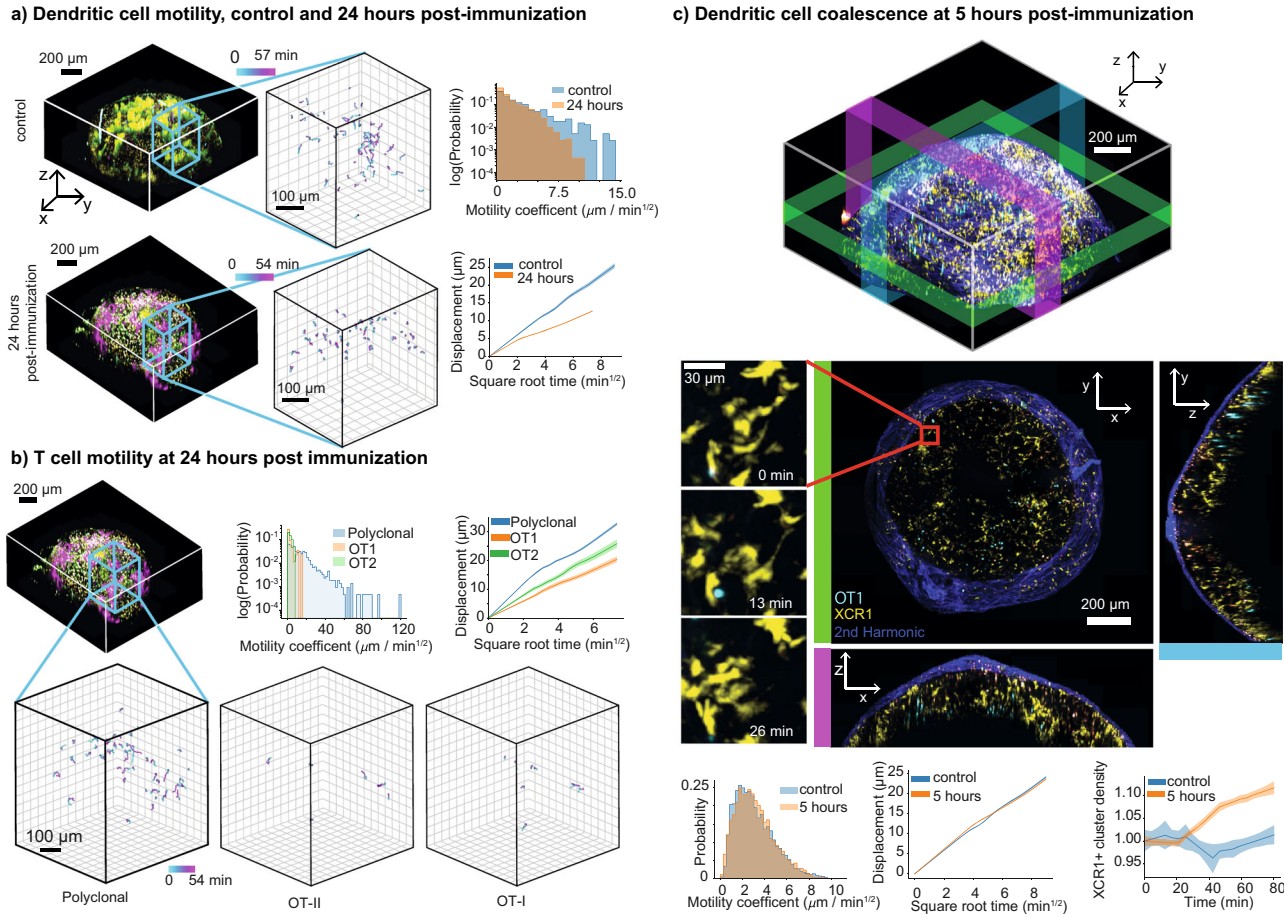

**Fig. 3 Immune response under physiological conditions. a** Distinct changes in global behavior of antigen-presenting cells as measured by XCR1+ dendritic cell motility 24 h after immunization show the cell behavioral correlates of developing immune responses. (Left) Tracks of motility in control and 24 h post immunization, (right top) log histograms of motility coefficients, and (right bottom) displacement vs square root of time show that dendritic cells switch from faster random walk behavior in the control (i.e., straight line in bottom right plot) to slower, confined motility 24 h post immunization. **b** T cell motility at 24 h post immunization. (Top middle) log histograms of OT1, OT2, polyclonal T cells. (Top right) Displacement vs square root of time plots. (Bottom) tracks of T cell motility. **c** Dendritic cell clustering can be visualized and quantified on the whole lymph node level. (Top) 3D view with colored bars marking areas shown in 2D projections below. XY, YZ, XZ projections with zoomed-in area show an example of dendritic cell cluster forming over 26 min. (Bottom) Histograms of dendritic cell motility at 5 h post infection vs control, mean displacement vs square root of time, mean normalized density over time in 5 h post infection vs control dataset show that formation of dendritic cell clusters can be detected on the timescale of 1 h, but without any detectable change in dendritic cell motility. Error bars on all plots show 95% confidence intervals.

organelles or fluorescent beads. Other samples will also require a means of building a map of the sample surface. Though this work uses second harmonic generation (SHG) signal at the sample surface, reflected visible light might be better suited for this purpose. This process could also be automated to improve imaging speed.

Although scattering of excitation light is likely the largest factor responsible for the drop in fluorescence with depth, absorption of emission light may also play a role, especially when imaging deeper into the sample. The fact that far-red fluorophores can be seen at greater depths than those in the visible spectrum supports this possibility (e.g., eFluor670 cells in Fig. 2b). Since the neural network makes no distinction between a loss of fluorescence from scattered excitation light and one from absorbed emission light, it is possible that the network learns to compensate for some combination of the two. Compensating for absorbed excitation light would imply that fluorescence emission and photobleaching increase at greater depths (which anecdotally seemed to be the case). It is also possible that the sample does not have a spatially uniform scattering potential, but that the neural network learns to implicitly predict and compensate.

There are many areas in which LAMI could be improved. The biggest issue in delivering the correct amount of power to each point in intravital imaging of lymph nodes was the map of the sample surface becoming outdated as the sample drifted over time. To combat this, we employed both a drift correction algorithm and periodically recreated the surface in between time points based on the most recent imaging data. We note that our system used a modified multiphoton system not explicitly designed for this purpose, and building a system from scratch with better hardware synchronization between scanning mirrors, focus, and excitation power would increase temporal resolution several fold and lessen the impact of temporal drift. Using state-of-the art image denoising methods[24] would also allow for faster scanning.

The maximum depth of LAMI in our experiments was limited by the maximum excitation laser power that could be delivered. A more powerful excitation laser could push this limit deeper, or using three-photon, rather than two-photon excitation. Another improvement to depth could be made by coupling adaptive illumination with AO. Incorporating AO could lessen the loss in resolution with depth and potentially restore diffraction-limited

resolution deep in the sample. Combining ideas form LAMI with AO could be especially powerful. One limitation of AO in deep tissue MPM is the need for feedback from fluorescent sources to pre-compensate for scattering[5,25,26], making the achieved correction dependent on the brightness and distribution of the fluorescent source being imaged. We have demonstrated in this work that it is possible to predict the appropriate excitation amplitude from sample shape alone. We speculate that a similar correction might be possible for the phase of excitation light, since scattering is caused by inhomogeneities in refractive index, and the largest change in refractive index seen by the excitation wavefront is likely to be at the surface of the sample when it passes from water into tissue. Deterministic corrections based on the shape of the sample surface have indeed shown to improve resolution in cleared tissue[27], and the additional flexibility of a neural network could provide room for further improvement.

In contrast to contemporary techniques based on deep learning[19], the neural network we employ is simple and shallow (1 hidden layer with 200 hidden units). Adding layers did not increase the performance of this network on a test set. We believe this is a consequence of the relatively small training set sizes we used ($10^4$–$10^5$ examples). Larger and more diverse training sets and larger networks would likely improve performance and potentially allow for additional output predictions such as wavefront corrections.

In conclusion, LAMI is a powerful technique for adaptive illumination in multiphoton imaging, with the potential for opening a range of biological investigations. We were able to implement it on an existing two-photon microscope using only an Arduino-like programmable micro-controller and a small op-amp circuit for less than $50. A tutorial on how to implement LAMI using exclusively open-source hardware and software can be found on Zenodo[28].

## Methods

**Quantifying the increased volume imaged with LAMI.** In order to understand the increases in volume provided by LAMI, we must first consider the problem of signal decay in MPM, the commonly used techniques for addressing this problems, and the unique challenges that arise when applying these techniques to curved samples.

*Challenge 1: non-exponential decay profile.* The literature reports that two-photon fluorescence decays exponentially with depth at constant power, and thus requires an exponential increase of power with constant depth in order to compensate and achieve uniform signal[1]. A simple geometric argument demonstrates why this is not true of curved samples. The exponential increase in attenuation is based upon the assumption that the path lengths through the sample of excitation rays increase linearly with depth. However, in a curved sample this is not the case. Specifically, in the case of convex samples like lymph nodes, the distance traveled through the sample by marginal rays increases sublinearly as a function of the distance focused into the sample (Supplementary Fig. S5a, b). As a result, the required power to compensate and achieve uniform fluorescence must be sub-exponential with depth (Supplementary Fig. S5c).

*Challenge 2: decay profile must be relative to top of sample.* In multiphoton systems where an arbitrary function (i.e., not just an exponential) can be set to increase power with depth, the power increase profiles are usually a function of the microscopes Z-axis. For a curved sample, this means that the power profile will not be applied from the top of the sample itself. Thus, in order to properly apply a decay profile, the microscope must incorporate some knowledge of the position of the top sample and offset the decay profiles appropriately.

*Challenge 3: functional form of decay profile changes across curved samples.* Even with the ability to apply arbitrary offsets depending on the location of the top of the sample, the function mapping depth to fluorescence decay can change across the sample depending on the local curvature being imaged through. To be able to image a curved lymph node in full, one must know an ensemble of such profiles, such that the appropriate one can be applied based on the local shape.

*Estimating the increase in volume using LAMI.* Before getting into the details of the calculations, an important point must be clarified: For a given object in the sample, there are a range of laser powers that might appear to be acceptable. That is,

anything above the threshold where it becomes visible and below the threshold at which visible heat damage occurs. However, photobleaching and photodamage are occurring well below the upper threshold where the sample can be clearly seen burning. Thus, our criterion is not to end up anywhere in this range, but rather to be at its very bottom: generating enough emission light for visualization and analysis with the minimal possible excitation power.

Supplementary Fig. S6a shows a comparison of various potential strategies for spatially modulating illumination in a popliteal lymph node, our calculations for the volume that each would be able to image, and the parameter values used in those calculations. The 3D volume of illumination power predictions made by LAMI is used as the target value for excitation power at each point.

The top row shows the simplest strategy: constant illumination. With this strategy, a small strip on the upper portion on the lymph node, where the required power is approximately constant, can be made visible. This strip does not extend to the lower portion because shadowing of half of the excitation light requires greater power here. It is not possible to image larger areas deeper in the lymph node without overexposing adjacent areas. We model this as a spherical shell with a 25 µm thickness. We multiply the resultant volume by a factor of $\frac{1}{4}$ as a rough estimate of the fraction of the hemispherical shell not affected by this shadowing.

Most multiphoton systems are equipped with an ability to modulate laser power with depth. In many cases, this consists of setting a decay constant to modulate power according to an exponential function. Flat samples often have such an exponential profile with depth, but curved samples such as lymph nodes do not (as shown theoretically in Supplementary Fig. S6). Based on the empirical fit in Supplementary Fig. S6b, we conclude that such a strategy only works up to 140 µm deep, and thus set the $h$ parameter for this method equal to 140.

Many multiphoton systems are not limited to only an exponential increase, but can increase power with depth according to an arbitrary, user-specified function. In this case, the microscope can image up to the depth limit according to the maximum laser power, which on our system is ~300 µm. Thus, we set the $h$ parameter for this method equal to 300.

In either case, a typical multiphoton system will do this modulation as a function of the coordinate of the position of the Z-drive. Thus, these profiles only remain valid for XY shifts over which the top of the sample has not changed significantly in Z. The radius of this shift was used as the value of $r$ in our in rows 2 and 4 of Supplementary Fig. S6a. To estimate it, we plotted a series of XZ profiles of the ground truth excitation at different lateral shifts (Supplementary Fig. S6). These profiles began at the Z coordinate of the top of the sample, and did not shift as the top of the sample changed (because MPMs without the ability to modulate power in X, Y, and Z during a scan, as achieved without TR-SLM, cannot do this). The corresponding line profiles stay constant for up to 250 µm when imaging the central part of the lymph node, giving us an estimate of 125 µm for the $r$ parameter. This is a best case estimate, because the required profiles are only relatively constant with Z in the central, flatter part of the lymph node. In other areas of the lymph node (which researchers are often interested in imaging, since relevant biology can be quite location specific in these structured organs), these profiles vary much more quickly, staying constant for no more than 100 µm (giving $r = 50$).

Imaging volumes larger than the previous cases require the ability not just to modulate power along the Z-axis, but also to (1) modulate power as a function of X, Y, and Z, and (2) have a map of the top surface of the sample. The latter is necessary so that the function for increasing laser power can be offset relative to the surface of the sample, rather than being a function of the microscope Z-drive's global coordinate space. The former is necessary because the Z coordinate of the surface of the sample can change substantially over a single FoV, so offsetting this function within a single FoV requires the ability to apply different excitation profiles in Z at different XY locations.

Using these two technologies in concert, the illumination system is no longer limited by the shift of the sample surface relative to the Z coordinate of the microscope. That is, the function for increasing power with depth can now be applied with an arbitrary Z offset. In this regime, the lateral extent of what can be imaged is now limited by the distance over which shifted versions of that function remain valid. Closer to curved edges of the sample, the functions shape must change to avoid over-illuminating the sample. To estimate this value, we plotted a series of XZ profiles of the ground truth excitation at different lateral shifts, starting from the top of the sample at the given lateral location (Supplementary Fig. S6d). From these, we estimate a value of 500 µm, and thus set $r$ equal to 250 in row 4 of Supplementary Fig. S6a.

To realize the full potential of the multiphoton microscope, we must not only be able to apply an arbitrary amount of excitation power in X, Y, and Z, but also have a robust method for both learning the function mapping shape to power and applying it in real time. The former is accomplished by the TR-SLM, and the latter by LAMI. Using both of these together the full volume of the lymph node (up to the excitation power limit) can be imaged, applying no more than the minimum necessary power. We calculate this as a spherical shell with an outer radius of 510 µm and an inner radius given by the depth that can be imaged with the laser at maximum power: (510 minus 300).

**Microscope.** All imaging was performed on a custom-built two-photon microscope (with 20 × 1.05 NA water immersion objective) equipped with two Ti:sapphire lasers, one MaiTai (Spectra-Physics) and one Chameleon (Coherent). The

former was tuned to 810 nm and the latter was tuned to 890 nm in order to provide a good combination of incident power and excitation for the set of fluorophores used. The microscope had six photomultiplier tube detectors in different bands throughout the visible spectrum, giving six-channel images. All data were collected using Micro-Magellan[29] software to control the Prior Proscan II XY stage and two Z drives, a ND72Z2LAQ PIFOC Objective Scanning System with a 2000 μm range, which was used to translate the focus during data collection, and a custom built stepper-motor based Z-drive, which was used to re-position the sample due to drift in between successive time points. All Z-stacks were collected with 4 μm spacing was used between successive planes.

**Spatial light modulator.** Because the appropriate excitation power varies as a function of X, Y, and Z, we need to modulate laser intensity over all of three dimensions. However, typical two-photon microscopes are equipped to only modulate intensity over Z—by changing the laser intensity between different focal planes. Thus, a custom TR-SLM was built to provide the ability to pattern illumination across a single XY focal plane. By applying different 2D patterns at each focal plane, the laser intensity could be modulated across X, Y, and Z. This TR-SLM takes advantage of the scanning nature of MPM—that is, the final image is built up pixel-by-pixel in a raster scanning pattern. This scanning pattern is physically created inside of the microscope by the changing the angle of deflection of two scanning mirrors. One of these mirrors operates in resonant scanning mode, oscillating back and forth with sinusoidal dynamics to control X position within the image. The second mirror is a galvanometer which operates with linear dynamics to control the Y position within the image. Both mirrors are controlled by a custom built controller box (Sutter Instruments), which outputs TTL signals corresponding to completion of a single line and completion of a full frame (line-sync and frame-sync, respectively).

The basic operation of the TR-SLM is to take these TTL signals as input, determine where in the FoV is currently being scanned, and apply appropriate modulation to the excitation laser based on a pre-loaded pattern. A circuit diagram for the TR-SLM can be seen in Supplementary Fig. S3. The TR-SLM is built from a Teensy 3.2 (a programmable micro-controller) using the Arduino IDE. It connects to the controlling computer via USB, through which a low-resolution (8 × 8) XY modulation pattern is pre-loaded via serial communication. The TR-SLM is also connected to the mirror controller's frame-sync and line-sync TTL signals. Each time one of these signals is received, an interrupt fires, which initiates a corresponding timer. In between interrupts, current scanning position in X and Y is determined based on the elapsed time on these timers (using the appropriate inverse cosine mapping for the resonant scanner). The laser modulation is then determined for that point by bilinear interpolation of the low-resolution pattern. This ensures the ability to apply a smooth gradient of excitation across the field rather than a discretized one determined by the resolution of the supplied pattern.

The excitation laser's amplitude is controlled by an EOM, which takes a logic-level input of 0–1.2 V (where 0 V is off and 1.2 V is full power). The EOM's input is controlled by the Teensy's onboard digital-to-analog converter (DAC) via a voltage divider and voltage buffer circuit. The DAC output is put through a voltage divider to lower the logic level from 3.3 to 1.2 V in order to utilize the full 12-bit analog control, and the signal is then run through a LM6142 rail-to-rail operational amplifier in a buffer configuration to isolate the DAC output from any downstream current-draw effects.

To validate the performance of the TR-SLM, a uniform fluorescent plastic slide was imaged with different patterns projected onto it (Supplementary Fig. S2). Supplementary Fig. S2a shows a checkerboard pattern, which is not a realistic pattern that would be projected in a lymph node, but demonstrates the resolution capabilities of the TR-SLM. Along the vertical axis, the pattern can be precisely specified on a pixel-by-pixel basis. However, the pattern is blurred along the horizontal direction, resulting from the average of many images, each with a noisy pattern along that dimension due to the resonant scanning mirror moving along the horizontal axis faster than the vertical axis. The fundamental limitation is the clock speed of the Teensy, which limits how fast the voltage to the EOM that modulates the excitation laser can be updated. However, in practice, this noise is not a problem because the excitation power needed is a smoothly varying function, and thus a more realistic pattern for imaging into a sample is a gradient pattern (Supplementary Fig. S2c).

**Imaging experiment setup.** The popliteal lymph node was surgically exposed in an anesthetized mouse. Because of the geometry of our surgical setup, only one half of the popliteal lymph node was visible (i.e., the axis running from top of cortex to medulla was perpendicular to the optical axis). Although we were able to image this half of the lymph node, to get a better view of the whole cortical side of the lymph node, we had to cut the afferent lymphatic, so that the lymph node could be reoriented with its cortex facing the objective lens. The efferent lymphatic and blood vessels were left intact. We note that a better surgical technique might be able to circumvent this limitation.

To start the experiment, the microscope was focused to a point on the top of the lymph node cortex using minimal excitation power and the signal visible from SHG. Micro-Magellan's explore mode was then used to rapidly map the cortex of the lymph node using a low excitation power, and interpolation points were marked on collagen signal from the SHG image. This surface was used not only to

predict the modulated excitation power, but also to guide data acquisition. Using Micro-Magellan's distance from surface 3D acquisition mode, only data within the strip of volume ranging from 10 μm above the lymph node cortex to 300 μm were acquired, rather than the cuboidal volume bounding this volume. This avoided wasting time imaging areas that were either not part of the lymph node, or so deep in it that they are below the depth limit of two-photon microscopy. Over time the volume being imaged tended to drift. This was partially compensated for by using the drift correction algorithm described below. However, we limited the use of this algorithm to drift in the Z direction where drift tended to be the most extreme (presumably because of thermal effects or the swelling of the lymph node itself). For XY drift, or for Z drift if the algorithm did not correct, we periodically paused acquisition and marked new interpolation points on the cortex of the lymph node, in order to update both the physical area being imaged, and the automated control of the excitation laser.

**Image denoising.** All data were denoised using spatiotemporal rank filtering[29]. Two full scans of each FoV were collected at each focal plane before being fed into a 3 × 3 spatial extent rank filter. Because of the computational load of performing all the sorting operations associated with this filtering at runtime, a computer with a AMD RYZEN 7 1800X 8-Core 3.6 GHz processor was used for data collection, and the filtering operations were parallelized over all cores. In addition, the final reverse-rank filtering step was done offline to save CPU cycles during acquisition. Before processing data, an additional filtering step using a 2D Gaussian with a 2-pixel sigma kernel was applied to each 2D slice to improve signal-to-noise on downstream tasks. We note that while spatio-temporal rank filtering was designed specifically for the task of cell detection applied here, there may be room for further improvement of real-time denoising (and thus lower doses of excitation light) strategies based on deep learning[24].

**Ray optics spherical excitation model.** On the way to developing standard candle calibration, we experimented with a simulation framework in which lymph nodes were modeled as spheres with homogeneous scattering potential. This framework had several disadvantages, as detailed below. However, it was useful as a starting point for calculating the random excitation powers that were applied to generate the standard candle training data (even though this may not have been absolutely necessary for generating random excitation). It is also useful to understand why it is difficult to accurately make a physics-based model of this problem, and why machine learning is especially useful. The details of this model are described below.

For a single ray propagating through tissue the proportion of photons that remain unscattered and the two-photon fluorescence intensity decay exponentially with depth[1]: $F = P_0 e^{-\frac{2z}{l_s}}$, where $F$ is the two-photon fluorescence excitation, $z$ is the distance of propagation, $P_0$ is the incident power, $l_s$ is the "mean free path" for a given tissue at a given wavelength, which measures the average distance between scattering events.

We assume a beam with a Gaussian profile at the back focal plane of an objective lens, which implies that the amplitude and intensity of the cross-sectional profile of the focusing beam are also both Gaussian. We also assume the contribution from photons that are multiply scattered back to the focal point is negligible and that scattered light does not contribute to the two-photon excitation at the focal point. Using a geometric optics model with these assumptions, the attenuation of each ray propagating toward the focal point can be considered separately. Thus, we can calculate the amount of fluorescence emission at the focal point by numerically integrating over all rays in the numerical aperture of the objective, with a known tissue geometry and scattering mean free path (Supplementary Fig. S4a). Supplementary Fig. S4b shows the output of such a simulation for a spherical lymph node of a given size. Relative excitation power is that factor by which input power would need to be increased to yield the same fluorescence as if there were no scattering. It is parameterized by the vertical distance from the focal point to the lymph node surface, and the normal angle at that surface.

This model suffers from three main drawbacks that preclude its usefulness for predicting excitation in real time: model calibration, model mismatch, and speed.

First, in order to calibrate such a model, two difficult to measure physical parameters of the microscope and the sample must be estimated: the complex field of excitation light in the objective pupil plane and the scattering mean free path of the sample. The model is sensitive to miscalibrations of the former, because different angles travel through different lengths of tissue in the sample, so an overfilling or underfilling of the objective back aperture can have a major influence on the amount of light that reaches the focal point. Estimating this likely requires some kind of PSF measurement along with a phase retrieval algorithm (though our model instead used an estimate of a Gaussian profile with zero phase). The second needs to be measured empirically, as such are not comprehensively available for different wavelengths and different tissue types in literature.

Second, even if the model can be calibrated, it will only work if the model captures all the relevant physics of the problem. Thus, if wave-optical effects play an important role here (which they might, with a coherent excitation source such as laser), the model will fail to account for this. The model also assumes a homogeneous scattering potential throughout the lymph node, which we do not necessarily know to be true. In contrast, neural networks are a much more flexible

class of models, with the ability to fit many different types of functions without being hindered by model mismatch.

Third, and most importantly, such a model is very computationally costly. It must integrate over the full 2D distribution of rays within the microscope's numerical aperture, calculating the propagation distance through the sample for each one. Our implementation of this took ~1 s per focal point, and 64 such calculations must be done for each field view, which is acquired in 60 ms. Such a model would need to be sped up 1000× in order to be applied in real time. Because of this, the implementation used to generate the data in our figure had to be pre-computed, and thus could not know the actual shape of the sample, adding another source of model mismatch and potentially explaining the suboptimal performance. The neural network model, in contrast, could be evaluated in less than 1 ms, and could thus be applied in real time without extensive computational optimization.

**Standard candle calibration.** The training data for standard calibration were collected by imaging an inguinal lymph node ex vivo, which had previously been seeded with $2 \times 10^6$ lymphocytes from a Ubiquitin-GFP mouse and $2 \times 10^6$ lymphocytes from a B6, which had been labeled in vitro with eFluor670 (e670). Each population was used as a standard candle for one of the two excitation lasers on the system, which had their wavelengths tuned to 810 and 890 nm. Two separate images were recorded, one with each laser on. The lymph node was imaged by tiling multiple Z-stacks in XY to cover the full 3D volume. Each Z-stack was imaged with power determined using the output of the spherical model described above, multiplied by a randomizing factor drawn from a uniform distribution between 0.5 and 2. These randomly distributed brightness data were then fed into the cell segmentation and identification pipeline described below. The mean brightness was taken for all voxels within each segmented region as the brightness of the standard candle. The standard candle's spatial location was used to determine the EOM voltage applied at that point in space, its location in the XY FoV, and a set of statistics to serve as effective descriptors of the physics of light scattering and emission light absorption.

The physical parameters were computed by measuring 12 distances from the focal point of the standard candle to the top of the interpolation marking the cortex of the lymph node (Fig. 1 in main text). All 12 distances were measured along directions that had the same angle of inclination to the optical axis ($\phi$), with equally spaced rotations about the optical axis. Taken in its raw form, each element of this feature vector is associated with a specific absolute direction in the coordinate space of the microscope. The microscope should be approximately rotationally symmetric about its optical axis. We do not want the machine learning model to have to learn this symmetry from data, because it would needlessly increase the amount of training data needed. Thus, we explicitly build in this assumption by binning all distances into a histogram. We use nonlinearly spaced bin edges for this histogram, based on the intuition that scattering follows exponential dynamics with propagation distance, so relative difference in short distances of propagation are more significant than those same differences at long distances. The bin edges of this histogram were calculated by taking 13 equally spaced points from zero to one, and putting them through the transformation $f(x) = (x^{1.5})(350\,\mu m)$, where 350 μm is the propagation distance beyond which we do not expect excitation light to yield any fluorescence excitation.

Standard candle brightness, location in XY FoV, and the physical parameter vector were concatenated into a single feature vector. Each of these feature vectors corresponded to one standard candle cell and was associated with a scalar that stored the voltage of the EOM used to image that standard candle. The total number of these pairs was 4000 for the GFP standard candles and 14,000 for the e670 standard candles. We standardized all feature vectors by subtracting their element-wise mean and dividing by their element-wise standard deviation. We then trained a fully connected neural network with one 200-unit hidden layer and a single scalar output. The network was trained using the Adam optimizer, dropout with probability 0.5 at training, and a batch size of 1000. Training was continued until the loss on the validation set ceased to decrease.

The output of this network is the voltage on a particular EOM. Because the goal of this network is to deliver the right amount of excitation power, as opposed to voltage, we converted this voltage into an estimate of relative excitation power (in arbitrary units) before feeding it into a squared error loss function. We measured the function relating EOM voltage to incident power empirically by placing a laser power meter at the focal plane of the objective lens and measuring the incident power under several different voltages. We found this curve to be well approximated by a sinusoid, so we fit the parameters of this sinusoid and used it directly in the loss function.

We experimented with several different architectures before finding the one that worked best with our data. Neither adding additional hidden layers, nor increasing the width of the existing hidden layer beyond 200 improved performance. The best performing value of $\phi$ (the angle of inclination to the optical axis) was 20°. Neither other angles, nor using multiple angles improved performance on the validation set. This was somewhat surprising, as we would have expected more information about the local geometry to improve prediction. We suspect that this might be the case with a larger training set.

**Using standard candle calibration to control laser power.** On later experiments, we loaded the trained weights of the network, computed its output for 64 points in

an $8 \times 8$ grid for each XY image, and sent these values to the TR-SLM through serial communication. The element of the vector corresponding to standard candle brightness must be chosen manually, and can be thought of a $z$-score of the distribution of brightness in the training set (since the training set was standardized prior to training). For example, picking a value of 0 means the network will provide the right laser power to achieve the mean brightness in the training set. Picking a value of −1 means it will aim for a brightness 1 standard deviation below the mean value of the training set.

To calculate the physical parameter feature vector, we computed the interpolation of the lymph node surface as described previously. This interpolation yields a function of the form $z(x, y)$, where there is a single $z$ coordinate for every XY position (unless the XY position is outside the convex hull of the XY coordinates of all points, in which case it is undefined). To avoid having to repeatedly recalculate this function, it is evaluated automatically over a grid of XY test points and cached in RAM by Micro-Magellan. In order to fill out the physical parameter feature vector, we must calculate the distance from an XYZ location inside the lymph node to its intersection with the interpolated surface. We measure this distance numerically, using a binary search algorithm. This algorithm starts with a value larger than any distance we expect to measure (i.e., 2000 μm), tests whether the Z value for this XY position is above the surface interpolation or undefined (which means it is outside the lymph node), halves the search space, and repeats this test until the distance is within some tolerance (we used μm). These calculations were all handled on a separate thread from acquisition so that they could be pre-computed and not slow down acquisition. We note that our strategy of sending each pattern out as a serial command certainly prevents the system from running as fast as it might otherwise be able. Sending out many such patterns at once and relying on a system that uses hardware TTL triggering should dramatically increase the temporal resolution of this technique.

**Validating standard candle calibrated excitation.** To validate the use of standard candle calibrated excitation, we transferred $2 \times 10^6$ GFP lymphocytes, $2 \times 10^6$ RFP lymphocytes, and $2 \times 10^6$ e670 lymphocytes and imaged its mediastinal lymph node ex vivo. We note that the mediastinal lymph node is quite different in size and shape than an inguinal lymph node. The lymph node was imaged with constant excitation, excitation predicted by the spherical ray optics model, and excitation predicted by the standard candle neural network (Fig. 2). The transferred lymphocytes included both T cells and B cells, meaning that there should be fluorescently labeled cells throughout the volume of the lymph node.

**Drift correction.** Focus drift, primarily in the Z direction, was present in all experiments at rates on the order of ~1 μm/min. This is unsurprising given the massive influx of cells to lymph nodes during inflammatory reactions. It was essential to compensate for this drift, because not doing so would lead to a mismatch between the coordinates of our interpolation marking the lymph node cortex and its actual location, which would in turn mean the automated excitation would be misapplied. To compensate for this drift, we designed a drift compensation algorithm that ran after each time point, and changed the Z-position of secondary Z focus drive (i.e., not one used to step through Z-stacks) after each time point. Estimates of drift were based on the SHG signal from the fibers in the lymph node cortex, which were a convenient choice because their contrast was not dependent on fluorescent labeling, and their spectral channel (violet) had relatively little cross-talk with other fluorophores. At each time point after the second, the cross-correlation of the 3D image in violet channel was taken with the corresponding image from the previous time point. The maximum of this function was taken within every 2D image corresponding to a single slice, and a cubic spline was fit to these maxima. The argmax of the resulting smooth curve was used to estimate the offset in Z between two successive time points with subpixel accuracy. This estimate was used to update an exponential moving average that estimated the rate of drift, so that both the existing drift from the previous time point could be corrected, and the expected future drift could be pre-compensated for. In practice, this algorithm worked well enough to stabilize the sample enough for the adaptive illumination to be correctly applied. Remaining drift in the imaging data was corrected computationally as described below.

**Image registration.** In order to conduct in vivo investigations, we must first address motion artifacts, which are an inescapable feature of intravital imaging and can compound when imaging large volumes over time. We thus develop a correction pipeline based on iterative optimization of maximum a posteriori (MAP) estimates of translations for image registration and stitching (Supplementary Figs. S7 and S8). These corrections enabled the recovery of stabilized timelapses in which cell movements can be clearly visualized and tracked (Supplementary Movies S5 and S6).

We identified three types of movement artifacts that occurred during intravital imaging. (1) Due to the mouse's breathing, there were periodic movements of successive images relative to one another within each Z-stack. These movements could be well approximated by motion within the XY plane, in part because of the geometry of the imaging setup, and in part because of the heavily anisotropic resolution of the imaging system, in which objects were blurred out along the Z-axis much more so than X and Y. (2) Individual Z-stacks were misaligned with

each other in X, Y, and Z. This seemed likely to be caused by physical movement of the sample as a result of some combination of thermally induced focus drift and biological changes leading to small tissue movements. (3) Global movements of the entire sample over time. All three remained to some degree even after experimental optimizations to improve the system stability and pre-heating the objective lens to minimize thermal drift.

To correct these artifacts, we used a three-stage procedure with each step corresponding to a type of movement artifact. Although cross-correlation is often the first choice for rigid image registration problems in the literature, it was found to be ineffective for solving two of the three of problems. Thus, we employed a more general framework, using iterative optimization to compute MAP estimates. This framework depends on the ability to transform and resample the image in a differentiable manner. As shown in Supplementary Fig. S5a, we can set up a general image registration problem that can be solved by numerical optimization by creating a parametric model for how pixels move relative to one another, resampling the raw image based on the current parameters of this model, and then computing a loss function that describes how well the alignment based on this transformation is. This paradigm enables us to solve general MAP estimation problems of the following form with iterative optimization:

$$\boldsymbol{\theta}^* = \arg\min_{\boldsymbol{\theta}} L(f_1(\boldsymbol{\theta}), f_2(\boldsymbol{\theta}), \ldots) + R(\boldsymbol{\theta})$$

Where $\boldsymbol{\theta}$ are the parameters to optimize, $f_n$ is the transformation and resampling of the $n$th subset of pixels, $L$ the loss function, and $R(\boldsymbol{\theta})$ is a regularization term for the parameters that allows incorporation of prior knowledge. We used the deep learning library TensorFlow to set up these optimization problems. This had the advantage of being able to automatically calculate the derivatives needed for optimization using built-in automatic differentiation capabilities. Often, these problems used extremely large amounts of RAM, because all image pixels were stored in memory when performing optimization. We were able to do this by using virtual machines on Google Cloud Platform with extremely large amounts of memory (>1 TB). However, we note that it would be possible to reduce the RAM requirements by downsampling the images, or more carefully coding the optimization models to only use relevant parts of the images rather than every pixel.

For the first correction, movements in XY for each Z-stack, each Z-stack was optimized separately. We observed that XY movements were almost always confined to a single z plane and that looking at an XZ or XY image of the stack, these movements were clearly visible as discontinuities along the Z-axis. Thus, we parameterized the model by a (number of Z-planes) × 2 vector, corresponding to an XY shift for each plane. For this correction, all channels except for the channel corresponding to SHG were used. The loss function was taken as the sum over all pixel-wise mean-squared differences between consecutive z-planes, normalized by the total squared intensity in the image (which was necessary to ensure that the learning rate of the optimization did not need to be adjusted to accommodate the total brightness of the Z-stack):

$$L(\boldsymbol{x}, \boldsymbol{y}) = \frac{\sum_{j=0}^{N-1} \sum_{x', y'} (I(x' + x_j, y' + y_j, z_j) - I(x' + x_{j+1}, y' + y_{j+1}, z_{j+1}))^2}{\sum_{x', y', z'} I(x', y', z')^2},$$

where $\boldsymbol{x}$ and $\boldsymbol{y}$ are vectors holding the translations at each slice, $j$ is the index of the z plane, $N$ is the total number of Z-planes, $I(x, y, z)$ is a pixel in the raw Z-stack, and $x'$, $y'$ are the coordinates of pixels in the raw image. The regularization in this problem was a quadratic penalty on the sizes of the translations (implicitly encoding a prior that these translations should be normally distributed about 0) multiplied by an empirically determined weighting factor:

$$R(\boldsymbol{x}, \boldsymbol{y}) = \lambda (\|\boldsymbol{x}\|_2^2 + \|\boldsymbol{y}\|_2^2)$$

The value of lambda used was $8 \times 10^{-3}$. The model was optimized using the Adam optimizer and a learning rate of 1. Optimization proceeded until the total loss had failed to decrease for ten iterations.

The second correction, fixing movements over time, was computationally much easier to solve, because the strong signal of the similarity between consecutive time points in channels made registration not especially difficult if the correct channels were used. For this reason, this correction did not require iterative optimization, and could instead be solved with cross-correlation alone. The 3D cross-correlation was taken between every consecutive two time points for each Z-stack. The location of the maximum value of each of these cross-correlations gave the optimal 3D translation between consecutive time points, and taking a cumulative sum of these pairwise shifts gave an absolute shift for each stack over time.

The third correction, finding the optimal stitching alignment between each Z-stack, was the most computationally challenging of these problems. This is because it has a relatively small amount of signal (i.e., the overlapping areas of each Z-stack, which was less than 10% of the total volume of each Z-stack). Furthermore, the signal in these areas was relatively weak, because it was most susceptible to photobleaching since it is exposed to excitation light multiple times at each point. Furthermore, since stacks were often taken a few minutes apart, the content in these overlapping regions often changed. Compensating for these difficulties not only required using the iterative optimization framework with an appropriate loss function that accounted for variations in image brightness and proper regularization, but also carefully choosing which channels to use registration based

on the presence of non-motile fiducial signals. Most of the datasets we collected had a channel with high endothelial venules, a large and immobile structure in the lymph node, fluorescently labeled, and these channels were often the most useful due to strong signal and lack of movement. We also found good performance by including the SHG channel that provided signal from the collagen fibers in the lymph node cortex. Finally, we noticed that autofluorescent cells were numerous and immobile throughout the lymph node. Because autofluorescence has a broad emission spectrum that appears across three to four channels at once, as opposed to the labeled structures that appear over only one to two, we were able to isolate the signal from these cells by taking the minimum pixel value over several channels.

As shown in Fig. S8, each Z-stack was parameterized by a three-element vector that corresponded to its X, Y, and Z shifts. The loss function was taken as the mean of all of the correlation coefficients of the pixels in the overlapping regions of every pair of adjacent Z-stacks. Correlation coefficients are a better choice of loss for this task than cross-correlation, because they better account for variations in image brightness[30]. Optimization was performed using Newton's method with a trust region constraint. Rather than performing optimization on each time point separately, all time points were averaged together and a single optimization was performed taking all information into account. This was possible because relative movements between stacks that differed by time point had already been corrected by cross-correlations in step 2. Because of the strong signal afforded by averaging multiple time points together, no regularization was needed.

**Cell identification—feature engineering**. We developed a machine learning pipeline for tracking cell locations over time based on their fluorescent labels (Supplementary Fig. S8). Automating this process was essential, as some datasets contain thousands of labeled cells at 20 different time points. Our pipeline enabled their detection across all datasets with the manual classification of no more than 500 cells for each time, a task that could be completed in a few hours of manual effort. Briefly, this pipeline consisted of two stages: a 3D segmentation algorithm to identify cell candidates, followed by a neural network that used hand-designed features (Supplementary Fig. S9) to classify each candidate as positive or negative for a given fluorescent tag. We used an active learning[20] framework to efficiently label training data for this classification network, which led to a 40× increase in the efficiency of data labeling compared to labeling examples at random (Supplementary Fig. S8).

Cells were detected in a two-stage pipeline that first utilized 3D segmentation to identify cell candidates, followed by machine learning to classify which of those cell candidates belonged to a population of interest. Candidate regions were generated using the segmentation algorithm (Supplementary Fig. S8) built into Imaris 7.6.5 (i.e., the "surfaces" module), which includes a filtering step to smooth the data, a local background subtraction step to account for variations in brightness, a thresholding step to generate segmented regions, and a splitting step, in which seed points of a certain size are generated, and segmented regions are split based on these seed points. Candidates were generated for each population of interest (i.e., each fluorescent label) through the ImarisXT Matlab interface. Next, each candidate region was "featurized" by computing a set of descriptive statistics about the pixels enclosed within it. By default, Imaris outputs a set of 97 such statistics for each candidate, including intensity means, standard deviations, minimums, maximums, as well as a number of morphological features. However, these features are specific neither to the biological nor technical context of the data, and we found them to be not effective in all cases for training high-quality classifiers. Thus, we engineered a set of additional features to better capture the variations that are useful for classifying cells.

First, we reasoned that since all spectral channels are collected simultaneously in two-photon microscopy, the ratios of intensity in different channels contain important information. Treating each set of spectral statistics (e.g., intensity means for different channels) as a six-dimensional vector (for a six-channel image), we subtracted the background pixel value for each channel, and normalized to unit length. This "spectral normalization" takes advantage of the fact that that intensity measurements for a given fluorescent object are all proportional to the excitation power delivered to the focal point, and thus it normalizes intensity statistics while preserving their ratio. It also creates an additional feature from the magnitude of the vector prior to normalization, which captures the brightness of the object irrespective of it spectral characteristics.

We also designed several feature classes based on the observation that one of the failure modes of the segmentation algorithm in the candidate generation step was that it often created a single region around a cell of interest along with a second cell in close contact to it that expressed a different fluorophore, but had spectral bleed-through into the channel on which the segmentation was run. Thus, intensity-weighted centers-of-mass (COMs) within each region would be expected to show greater variance among the different spectral channels compared to a surface that surrounds a structure single source of fluorescence intensity. This should hold true even if the two objects surrounded by a single surface shared emission in the same channels, as long as the spectral profile of the two objects differs. With this in mind, we computed features for all pairwise distances between the intensity-weighted COM for different channels, as well as the distance from each intensity-weighted COM to the non-intensity-weighted COM. With the same reasoning, we also added the correlation matrices containing the pairwise correlations between

channels for all pixels within each candidate region as features (Supplementary Fig. S5a).

Finally, to more directly address the issue of overlapping, spectrally dissimilar cells (which are often the most biologically interesting case), we designed an algorithm to identify subregions of pixels within each candidate region that has a spectrum that is most similar to a reference spectrum (i.e., the spectrum of the fluorophore of the cell of interest). This algorithm is based on the normalized cut segmentation algorithm[31]. However, unlike that algorithm, which is designed for use on grayscale images, and builds an adjacency matrix for all pixels based on a combination of their spatial and intensity differences, our algorithm segments regions based on differences in their spatial and spectral distances. This is accomplished by defining distances between each pair of pixels as: $d_{i,j} = \alpha \parallel \mathbf{r}_i, \mathbf{r}_j \parallel_2^2 + \beta \hat{\mathbf{s}}_i^T \hat{\mathbf{s}}_j$, where $\mathbf{r}_i$ and $\mathbf{r}_j$ are the spatial coordinates of the two pixels, $\hat{\mathbf{s}}_i$ and $\hat{\mathbf{s}}_j$ are their unit norm intensity vectors across all channels, and $\alpha$ and $\beta$ are tuning parameters. Then, an adjacency matrix can be constructed by defining the adjacency between pixel $i$ and pixel $j$ as $w_{i,j} = e^{-d_{i,j}}$. The spectral clustering method defined in[32] can then be used to break all pixels into distinct regions, and the normalized cut region of interest (NC-ROI) most similar to a given reference intensity can be used for further downstream processing. Using this method, a number of additional features were calculated for the pixels within each NC-ROI.

To validate that these features were in fact useful, we performed two types of analyses. First, we looked at which types candidates cells the classifier repeatedly failed to correctly classify. By running $k$-fold cross-validation on a ground truth set of labeled candidates, we were able to identify which candidates the classifier failed to correctly classify. Overlaying these results on principal component analysis plot of the spectral variation among the cells of interest (RFP-labeled T cells), we found that the misclassified cells were often spectral outliers as a result of spatial overlap with some other fluorescent structure (Supplementary Fig. S6, left). However, including the engineered features in this experiment dramatically reduced the misclassification of these cells.

Next we ran a bootstrap analysis to identify the most useful features. We performed regularization and variable selection via the elastic net procedure. Elastic net is a useful method for identifying a sparse subset of useful predictors in a dataset with correlated predictors.

For the dataset examined, the number of labeled T cells was significantly lower than the number of non-T cells (T cells: 204, non-T cells: 38,575). In order to keep a balanced training and test dataset, we partitioned the data so that the model performs training and testing on similar sample distributions. In particular, we performed 100 bootstrap resampling procedures from both T cell and non-T cell data of approximately equal sample size. The model obtained was further tested on a smaller test dataset with equal T cell and non-T cell ratio, set aside at the beginning of the procedure.

For each bootstrapped sample set, we further ran 1000 iterations of randomly picking test set/training set partitions. This step was performed in order to assess the stability and overall distribution of the lambda parameter picked for each model. Despite lambda being chosen through five-fold cross-validation procedure, it is still specific to the prior decided training/test set partition. By performing additional random partitions of the bootstrapped dataset, we can break this dependence. In addition, we can also look at the distribution of cross-validation error provided by the glmnet package, as well as the misclassification error from the test set. The final model was fitted using the lambda parameter with the lowest cross-validation and misclassification error for the bootstrapped sample set. We looked at the averaged probabilities across all the samples, as well the total number of times each predictor was chosen by the elastic net out of 100 bootstrap subsamples.

The final results of this analysis can be seen in Supplementary Fig. S6d. Many of the engineered features are among the strongest predictors, further validating their usefulness for this task.

**Cell identification—active learning**. Having developed a useful set of task-specific engineered features that can train high-quality classifiers with sufficient labeled training data, the last remaining piece of the pipeline is a scheme for generating labeled training data. Often, this can be the most time-consuming piece of developing a machine learning system. To alleviate this bottleneck, we drew from the field of active learning, a paradigm in which a classifier chooses which data points receive labels, allowing them to learn more efficiently by seeing more informative examples[20]. Specifically, we use the strategy of "uncertainty sampling,"[33] in which the classifier outputs a number between 0 and 1 for each example (with 0 being complete certainty of one class and 1 being complete certainty of the other), and the example with a value closest to 0.5 is then selected and sent to a human for labeling. This process is then repeated until enough training data are labeled to train a classifier that generalizes well to the remaining unlabeled data.

Applied specifically to the problem of classifying which candidate regions corresponded to cells, the workflow was a follows: after computing the engineered features for all candidate regions, we first labeled one example of a candidate that belong to the population and one that did not. These labels were used to train the classifier (a small, fully connected neural network with 12 hidden units). Because classification accuracy usually increased by averaging the predictions of multiple neural nets, three were trained in parallel and their predictions were averaged. When making final predictions of cell populations, 100 neural nets were averaged.

The neural net was trained in Matlab, and the labeling interface for selecting cells was built with Imaris for data visualization, and Matlab script on the backend that communicated interactively with Imaris through the ImarisXT interface. We periodically predicted the identities of all candidates, in order to identify particularly difficult examples even faster and manually label them.

Although well justified from a theoretical standpoint as a means to make exponential gains in data labeling efficiency on idealized problems[34], there remained the question of whether active learning had the same effect on this problem. To answer this, we generated a ground truth set of labels by carefully manually labeling every cell on a limited dataset of candidates for RFP-labeled T cells. We reviewed each cell multiple times to be sure that its label was not a false positive, and searched manually through the data volume to identify any false negatives. Next, one positive and one negative candidate were randomly selected and assigned labels. This labeled set was used to simulate the uncertainty sampling procedure, drawing labels from the ground truth set rather than a human labeller. The accuracy on the remaining unlabeled examples was used to assess performance. As the plot in Supplementary Fig. S6d demonstrates, uncertainty sampling vastly outperformed random sampling for this classification task.

**Statistics and reproducibility**. All data analysis was performed in using custom scripts written using tools from the Scientific Python stack[35]. For all displacement vs root time plots, curves were fit to the means of scatterplot data using locally weighted scatterplot smoothing using locally linear regression[36] using a tricubic or Gaussian weighting functions. Sigma and alpha parameters were tuned manually for each comparison to capture the major trends in the data while smoothing out noise. Error bars represent 95% confidence intervals derived from bootstrap resampling of data with 500 iterations. Cells were tracked over multiple time points using the Brownian motion tracking algorithm in Imaris 7.6.5. In all cases the number of tracks used for quantification overestimates the number of cells since some cells move in and out of the frame or have breaks in their tracks where the algorithm fails. The visualizations of individual tracks used random subsamples of the total number of tracks for visual clarity. Details about sample sizes in individual sub-figures follow:

Figure 2b: representative of two sets of visualized predictions on different lymph node shapes, and consistent with uniform signal to noise seen across all LAMI experiments.

Figure 2c, d: representative of five experiments on the same lymph node. Consistent with results seen across all subsequent imaging data.

Figure 3a: imaging data were quantified for one dataset, which was representative of three and two movies taken in two animals for control and 24 h post immunization respectively. In all, 3015 and 514 tracked cells were used to generate the quantification for control and 24 h post immunization respectively. Tracks shorter than 5 min were excluded.

Figure 3b: imaging data were quantified for one dataset, which was representative of three movies taken in two animals (two separate regions were recorded in one animal). In all 1071, 44, and 10 tracked cells were used to generate the quantification for polyclonal, OT1, and OT2 cells, respectively. Tracks shorter than 3 min were excluded.

Figure 3c: imaging data were quantified for one dataset, which was representative of five movies taken in three animals for 5 h post LPS condition, and three movies taken in two animals for control condition. In all, 3483 and 5332 tracked cells were used to generate the quantification for control and 5 h conditions, respectively. XCR1+ cluster density was computed by counting the number of other XCR1 cells within 100 μm of each detected XCR1+ cell divided by the total number of XCR1 cells detected at each time point. Tracks shorter than 5 min were excluded.

Supplementary Fig. S1: representative of >10 experiments.

Supplementary Fig. S10a,b: representative of five experiments. Supplementary Fig. S10c: same as Fig. 3a. Supplementary Fig. S8c: bar plots represent proportions and correspond to 697, 1161, 301, 600 (control) and 114, 800, 517, and 12 (24 h post) cells In the subcapsular, deep paracortex, paracortex, and B cell regions, respectively. Supplementary Fig. S10g, h: same as Fig. 3b. Supplementary Fig. S8i–l: same as Fig. 3a.

Supplementary Fig. S11a: 861, 133, 649 (control) and 229, 53, 9 (24 h) were used to generate each plot. Supplementary Fig. S9b–d: same as Fig. 3a.

**Mouse strains**. All mice were treated in accordance with the regulatory standards of the National Institutes of Health and American Association of Laboratory Animal Care and were approved by the UCSF Institution of Animal Care and Use Committee (IACUC approval: AN170208). All mice were purchased for acute use or maintained under specific pathogen-free conditions at the University of California, San Francisco Animal Barrier Facility. Mice of either sex ranging in age from 6 to 12 weeks were used for experimentation.

The following mice were purchased from the Jackson Laboratory or bred to a C57BL/6 background: TCRa knock-out, LCMV P14-specific transgenic mice (MMRRC stock no:37394-JAX), C57Bl/6J (stock no:000664), OT-1 (stock no:003831), OT-2 (stock no:004194), transgenic mice interbred with CD2-RFP or ubiquitin-GFP (stock no:004353), and XCR1-Venus mice.

**Mouse immune challenge**. OTI and OT2 cells were isolated from lymph nodes of mice. In addition, polyclonal (C57BL/6) or LCMV P14-specific CD8+ T-cells were isolated as negative controls. Selection was carried out with a negative selection EasySep mouse CD8+ or CD4+ isolation kit (STEMCELL Technologies, 19853 and 19852). If T cells did not have a transgenic reporter (CD2-RFP or ubiquitin-GFP), they were fluorescently labeled with one of eFluor670 (Thermo Fisher Scientific, 65-0840-85), Violet Proliferation Dye 450 (BD, 562158), or CMTMR (Thermo Fisher Scientific, C2927). Dyes were diluted 1000-fold and incubated with isolated cells for 10–15 min minutes in a 37 °C, 5% $CO_2$ incubator. T cells were injected retro-orbitally (r.o.) into Xcr1-Venus recipient mice in 50–100 μL volumes. The number of OT1 and OT2 transferred was $5 \times 10^4$ except for experiments conducted at 5 h post infection, where $5 \times 10^4$ cells were transferred in order to visualize more T cell–dendritic cell interactions; for each experiment, equal numbers of OT1 and OT2 cells were transferred. 1e6 control T-cells were transferred. Mice were given a 30 μL footpad injection containing 2.25 μg LPS (Sigma-Aldrich, L6529-1MG) and 20 μg OVA protein (Sigma, A5503-1G) 1–4 days after T-cell transfer; a 30 μL footpad injection of DPBS was used as a negative control to the infection model. To visualize high endothelial venules, in some imaging experiments, 15 μg Meca-79 Alexa Fluor 647 (Novus Biologicals, NB100-77673AF647) or Alexa Fluor 488 (Novus Biologicals, NB100-77673AF488) was transferred r.o. in a volume of 50 μL immediately before imaging.

**Reporting summary**. Further information on research design is available in the Nature Research Reporting Summary linked to this article.

## Data availability

Source data are provided with this paper on FigShare[37]. The authors declare that all other data supporting the findings of this study are available within the paper and its supplementary information files.

## Code availability

A streamlined Jupyter notebook that describes how to implement LAMI can be found on Zenodo[28].

All other code including data analysis code can be found on Zenodo[38].

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

## Acknowledgements

This project was funded by a Packard Fellowship award to L. W.; STROBE: A NSF Science & Technology Center; A Google Cloud research grant award to H. P./L. W.; an NSF Graduate Research Fellowship awarded to H. P.; and a Berkeley Institute for Data Science/UCSF Bakar Computational Health Sciences Institute Fellowship awarded to H. P. with support from the Koret Foundation, the Gordon and Betty Moore Foundation, and the Alfred P. Sloan Foundation to the University of California, Berkeley. The authors thank BIDS and its personnel for providing physical space and general logistical and technical support, Jonathan Shewchuk for helpful conversations during development, and Sandra Baker and Bob Pinkard for support over the course of the project.

## Author contributions

Conceived of idea of whole lymph node imaging: M.F.K. Conceived of and implemented LAMI: H.P. with guidance from L. W. Designed and built TR-SLM: H.P. and D.H.F. Automated cell detection and tracking: H.P., I.M., and A.F. Designed and performed experiments: H.P., H.B., A.M., E.R., K.H.H., T.S., and K.C. Analyzed data and made figures: H.P., L.W., and M.F.K. Supervised project: M.F.K. and L.W.

## Competing interests

The authors declare no competing interests.

**Additional information**

