## [Peer Review File · Nature Communications]

Reviewers' Comments:

Reviewer #1:

Remarks to the Author:

As noted in my original review, this manuscript describes an enhancement to 2 photon imaging that will be of considerable interest to those imaging within small, highly curved tissues and organs, and particularly for immunoimaging of cells in lymph nodes. The revised version includes helpful clarifications and more extended explanations. My only reservation is that this is a niche application that may be too specialized for the wide readership of Nature Communications.

My main previous criticism of the original manuscript was the claim of as much as three orders in magnitude increase in imaging volume achieved by their TR-SLM + LAMI system as compared to conventional 2P imaging. That seemed implausible, and indeed the authors now admit the numbers came from a 'back of the envelope' calculation! The present manuscript now describes a lesser, 2 orders of magnitude improvement in numbers of transferred cells (abstract) and a 10-100x larger imaging volume (line 235 and Fig S6). I still do not find these numbers to be well supported or explained and suspect they may still be overestimates.

The numbers of cells that were transferred are not by themselves, meaningful without knowing the respective numbers of cells actually visualized with good fidelity within the imaging field. A simple, direct comparison would be to image the same node with the TR-SLM turned off vs. using constant laser power or power modulated only as a function of z-depth.

Fig. S6 presents calculations comparing imaging volumes achieved by different imaging modalities. The results appear to depend crucially on some arbitrary threshold for defining 'visibility' as "areas that can be imaged with an approximately constant profile in Z". For example, the green ("visible") area in the column showing arbitrary increase with Z profile looks unduly constrained in the X axis. Fig. S6 is quite complex, and needs more explanation and justification of parameters.

Whatever, the numbers given for reduction in transferred cells (5×10^5 vs. 10^6) and increased imaging volume (2.6×10^8 vs. 1.4×10^7) are both closer to a single order of magnitude (about 20 fold) rather than two order of magnitude improvements. Clearly there is a worthwhile improvement, but from the data presented it remains difficult to know quite how much of an improvement

Reviewer #2:

Remarks to the Author:

Thank the authors for revising the manuscript! The quality of the manuscript is highly improved. The work deserves publishing after addressing some minor problems.

There are some minor problems in the revised version:

1. I did misunderstand the SLM. This SLM is called time-realized spatial light modulator.

However, the claim of 50 dollars is still wrong. As shown in Figure S3, the time-realized spatial light modulator consists of electronic circuits and EOM. The electronic circuits are within 50 dollars, but the EOM is expensive, for example, Thorlabs EOM, EO-AM-NR-C2, is \$2,678.25. In the meantime, many 2P microscope does have EOM, then their claim is right.

2. In the reply, the authors claimed, "It worked in inflammatory (when lymph nodes swell to >2x their original size) and steady state conditions." This is worrisome. The logic does not make sense. Why we need to change power? The reason is due to scattering in the nontransparent tissue. Thus, when the tissue swells to >2x, the density of the tissue is less, the scattering of the tissue is less. Notice that after swelling, there is more water in the tissue, while the absorption of the light is negligible at this wavelength, compared to scattering. If the scattering of the tissue is less, the optical power change will be different. As shown in the main text, "On subsequent experiments in different samples of the same type, the model can predict the excitation power required to achieve a desired level of detected fluorescence for each point in the sample based only on sample shape and XY position." The input of the system is only SHAPE and XY position. There is no input for tissue transparency. It

does not make sense the training model can automatically know the transparency of the tissue.

To make it clear. A figure is attached above. F1 is before swelling and F2 is after swelling. For position A1 and B1, the best powers are both P0. For position A2 and B2, the best powers are also both P1. The reason is that although there is more water in the tissue, the absorption of light at the wavelength for 2P GFP is negligible. (Btw, where is the information for the laser, wavelength, and microscope, etc?) However, as seen by the training model, B2 is deeper and further, it should predict B2 requires more power than A2. As shown in F1 and F2, L2 is longer than L1. The claim is very worrisome.

3. Figure 2b looks much better. If one carefully compares ray-optics model and adaptive model, it is hard to say ray-optics model is much worse.

In the center area:

The power used in adaptive model is always higher than ray-optics. One can see both at top and bottom area, the adaptive model is much brighter. What one expects is, the brightness at the top area are the same, while in some deeper area, the ray-optics model is not as accurate as adaptive model.

In the right area:

One can see some red cells in the ray-optics model, while they disappear in the adaptive model.

Reviewer #3:

Remarks to the Author:

In the revised manuscript and the response to review, several points are clarified. There are some remaining issues.

1, The authors claim that the machine learning model is much improved compared to the physical optics model (based on the surface profile and a constant decay coefficient). Although the lymph node has a heterogeneous structure, the scattering coefficient does not have any sharp variation within the typical imaging depth. As the authors claim the machine learning method is superior, I hope the authors can quantify how much the machine learning outperforms the simple physical optics model. Machine learning is a buzzword. In many important applications, it indeed outperforms other approaches. However, based on the nature of the application and the presented data, I am not convinced that it significantly outperforms physical optics models. Based on the acquired data, I hope the authors can quantify the improvement (is it 2x, 20x better or just 20% better?).

2, The authors claim that the presented method can compensate for the absorption of emission light (line 284). Is there any experimental data that can back up this claim? If so, please add this to SI. If not, please consider removing this claim.

Reviewer #1 (Remarks to the Author):

As noted in my original review, this manuscript describes an enhancement to 2 photon imaging that will be of considerable interest to those imaging within small, highly curved tissues and organs, and particularly for immunoimaging of cells in lymph nodes. The revised version includes helpful clarifications and more extended explanations. My only reservation is that this is a niche application that may be too specialized for the wide readership of Nature Communications.

My main previous criticism of the original manuscript was the claim of as much as three orders in magnitude increase in imaging volume achieved by their TR-SLM + LAMI system as compared to conventional 2P imaging. That seemed implausible, and indeed the authors now admit the numbers came from a 'back of the envelope' calculation! The present manuscript now describes a lesser, 2 orders of magnitude improvement in numbers of transferred cells (abstract) and a 10-100x larger imaging volume (line 235 and Fig S6). I still do not find these numbers to be well supported or explained and suspect they may still be overestimates.

In addition to the new figure (S6), there is a page and half of explanation for our calculation of imaging volume increase in the supplementary materials supporting and explaining our calculations. Did the reviewer possibly miss this? If not, can the reviewer point to a specific part of our methodology/assumption/calculation that needs improvement or what is missing? It is now specifically noted in the main text that the value of this estimate changes depending on what baseline is compared to, which depends both on subjective beliefs and what type of experiments are being performed. We've done our best to justify and detail our calculations, as well as note their limitations.

The numbers of cells that were transferred are not by themselves, meaningful without knowing the respective numbers of cells actually visualized with good fidelity within the imaging field. A simple, direct comparison would be to image the same node with the TR-SLM turned off vs. using constant laser power or power modulated only as a function of z-depth.

While we agree that this suggestion would make sense if we were using the number of cells as a proxy for the volume of tissue that can be imaged, this is not our motivation. As described at line 81, the number of cells is significant because it determines the type of immune response under observation. These numbers are meant to demonstrate to a practitioner that qualitatively, our technique opens an entire new regime of physiologically realistic experiments. A lymph node with 5×10^6 monoclonal T cells has entirely different biology than one with 5×10^4 cells. It is significant that using our technique we can do the latter and still observe enough cells to make statistically sound conclusions about cell behavior (Figure 3), because this means we can observe an entirely new regime of immunology.

Fig. S6 presents calculations comparing imaging volumes achieved by different imaging modalities. The results appear to depend crucially on some arbitrary threshold for defining ‘visibility’ as “areas that can be imaged with an approximately constant profile in Z”. For example, the green (“visible”) area in the column showing arbitrary increase with Z profile looks unduly constrained in the X axis. Fig. S6 is quite complex, and needs more explanation and justification of parameters.

Yes, we agree, the calculations are complex and depend on subjective thresholds. Because of this there is a page and half of explanation of all parameters, assumptions, and calculations in the supplementary materials, in which we have made this clear. We note that the black and green figure is meant only as a qualitative cartoon diagram to better describe these calculations (hence the lack of any scale bars).

As explained there:

Before getting into the details of the calculations, an important point must be clarified: For a given object in the sample, there is a range of laser powers that might appear to be acceptable. That is, anything above the threshold where it becomes visible and below the threshold at which visible heat damage occurs. However, photobleaching and photodamage are occurring well below the upper threshold where the sample can be clearly seen burning. Thus, our criteria is not to end up anywhere in this range, but rather to be at its very bottom: generating enough emission light for visualization and analysis with the minimal possible excitation power.

“visible” is not the best word choice and we’ve since changed to “correct power”, but regardless, we know of no good methods for spatially quantifying the ultimate metric we would like to measure, induced photodamage, without great experimental difficulty. Thus we must make assumptions about how much of an increase in incident power is “too much.” These are subjective (though we do derive them from data as described in the materials and methods). This is a difficult estimate to make, and any final value comes with a high degree of uncertainty as a result. We have noted this and made every effort to make our calculations thorough and transparent. If the reviewer disagrees or misunderstands any of our assumptions, we will happily revise them, but without further specificity it is unclear how we should do so.

Whatever, the numbers given for reduction in transferred cells (5×10^5 vs. 10^6) and increased imaging volume (2.6×10^8 vs. 1.4×10^7) are both closer to a single order of magnitude (about 20 fold) rather than two order of magnitude improvements.

The two references we cite transferred $5-10 \times 10^6$ and $2-4 \times 10^6$ cells (which is why we wrote $>10^6$). 2 orders of magnitude is thus a reasonable estimate for the number of cells.

On the subject of volume increases, it is not clear which excitation schemes we should be comparing LAMI to in order to estimate the changes in volumes. In our previous revision, different reviewers suggested different schemes: (one an exponential profile with Z, one an arbitrary profile with Z). This is why we provided several alternatives to compare to. 10-100x seems to be a reasonable range accounting for the many potential comparisons one might make.

Clearly there is a worthwhile improvement, but from the data presented it remains difficult to know quite how much of an improvement

It doesn't seem likely to us that there is a satisfying, definitive answer to the question of how much improvement our technique offers (beyond, as the reviewer states, that it is worthwhile), since 1) a major factor in this improvement is the reduction in photodamage, which is not something we can easily measure. 2) It probably depends a lot on what experiment you're actually doing (what cell type, where in the lymph node they localize, etc.). 3) It depends on subjective assumptions like what baseline excitation strategy you compare to, about which there isn't even a consensus among the three reviewers of this manuscript.

Nonetheless, our technique clearly enables a large increase in volume, more reproducible, higher-quality images, and new types of experiments. Since our last revision we have created a detailed tutorial showing how every part of this technique can be reproduced with open source software and hardware that we've created as part of a different project for facilitating reproducible microscopy. By streamlining the process of adopting our technique, we have given practitioners the tools to scrutinize our technique and its advantages themselves, which we feel is a far more productive route for characterizing its utility than debating subjective assumptions about theoretically idealized samples.

Reviewer #2 (Remarks to the Author):

Thank the authors for revising the manuscript! The quality of the manuscript is highly improved. The work deserves publishing after addressing some minor problems.

Thank you

There are some minor problems in the revised version:

1. I did misunderstand the SLM. This SLM is called time-realized spatial light modulator. However, the claim of 50 dollars is still wrong. As shown in Figure S3, the time-realized spatial light modulator consists of electronic circuits and EOM. The electronic circuits are within 50 dollars, but the EOM is expensive, for example, Thorlabs EOM, EO-AM-NR-C2, is \$2,678.25. In the meantime, many 2P microscope does have EOM, then their claim is right.

Thank you for helping us to clarify this. As we state in the text, “We describe a simple hardware modification to an existing multiphoton microscope (costing <\$50)”. We don’t claim the full system costs less than \$50, just the modification. Nearly all modern 2P systems have some form of electronic power modulation control in the form of an EOM or AOM (ref), so modifying a 2P to include our TR-SLM shouldn’t require purchasing an EOM. We’ve clarified this in the text.

2. In the reply, the authors claimed, “It worked in inflammatory (when lymph nodes swell to >2x their original size) and steady state conditions.” This is worrisome. The logic does not make sense. Why we need to change power? The reason is due to scattering in the nontransparent tissue. Thus, when the tissue swells to >2x, the density of the tissue is less, the scattering of the tissue is less. Notice that after swelling, there is more water in the tissue, while the absorption of the light is negligible at this wavelength, compared to scattering. If the scattering of the tissue is less, the optical power change will be different. As shown in the main text, “On subsequent experiments in different samples of the same type, the model can predict the excitation power required to achieve a desired level of detected fluorescence for each point in the sample based only on sample shape and XY position.” The input of the system is only SHAPE and XY position. There is no input for tissue transparency. It does not make sense the training model can automatically know the transparency of the tissue.

To make it clear. A figure is attached above. F1 is before swelling and F2 is after swelling. For position A1 and B1, the best powers are both P0. For position A2 and B2, the best powers are also both P1. The reason is that although there is more water in the tissue, the absorption of light at the wavelength for 2P GFP is negligible. (Btw, where is the information for the laser, wavelength, and microscope, etc?) However, as seen by the training model, B2 is deeper and further, it should predict B2 requires more power than A2.

As shown in F1 and F2, L2 is longer than L1. The claim is very worrisome.

Thank you for the explanation. We should have more specifically said “expansion” instead of “swelling”. This expansion is caused not by the retention of additional water, but by the influx and division of cells (ref). We only mentioned this to demonstrate that the model generalizes to different sizes/shapes/curvatures. We don't think it would generalize (without modification) to a sample with different optical properties, as we have already noted in line 105 of the main text.

Info about laser/wavelength/microscope is at line 115 of the supplement.

3. Figure 2b looks much better. If one carefully compares ray-optics model and adaptive model, it is hard to say ray-optics model is much worse.

In the center area:

The power used in adaptive model is always higher than ray-optics. One can see both at top and bottom area, the adaptive model is much brighter. What one expects is, the brightness at the top area are the same, while in some deeper area, the ray-optics model is not as accurate as adaptive model.

To clarify, we don't think the difference between the ray-optics model and neural network model is a massive one, but it is still a noticeable and consistent improvement. As we say in the manuscript, "Given these limitations, we found that a physics-based neural network was a better solution to this problem."

The analysis presented above, in which one sub area is cropped out for comparison, is statistically flawed. It is not a fair comparison to look at only a subset of the data to assess the models' suitability, especially when there is a known source of model mismatch. Consider the analogy of fitting a parabolic point cloud with a linear model. If one only looks at a subset of data points in a relatively flat area of the parabola, a straight line can fit the data well. However, it cannot fit all the data points along the length of the parabola nearly as well.

The same is true of the ray optics model. Like fitting a parabola with a line, we know that it is inherently mismatched to the problem at hand, because it was not computed using true sample shape (a fact that is now more clearly described in the text). The full image from which the above is cropped shows that in other regions, particularly on the lymph node sides, the two models work equally well. If we were to tune the parameters to fit the center region better, the predictions in these areas where it currently works well would suffer (just as fitting a line to the bottom part of a parabola yields a poor fit on the sides). The fact that the ray optics model cannot image all areas at once with the proper amount of illumination power is one of the main reasons why it is inferior to the neural network model.

To make this figure, we had to pick a way to make a fair comparison between models. The places where the illumination is matched between the two models is partially due to our (somewhat arbitrary) decision to match the powers here for comparison purposes. This can be seen in the blue 2nd harmonic signal at the top of both samples, which is almost identical for the top 20 μm of the two illumination strategies. The areas of under-illumination occur where the ray optics model is mismatched to the sample. If the reviewer is curious to better understand the differences, we encourage them to look through the supplementary materials, where the datasets used to make these figures are available.

In the right area:

One can see some red cells in the ray-optics model, while they disappear in the adaptive model.

Good observation. Because we collected these images sequentially on a live sample, and because lymphocytes traffic very fast around lymph nodes, we don't expect the cells to be in the same position from image to image. This has now been clarified in the text. For the purposes of this experiment, it doesn't matter that we image the exact same cells, since the cells are homogeneously labelled and we can see the broad scale changes in brightness over larger regions of the lymph node regardless of their movement. We also note that there are at least 2 red cells in the bottom part of the bottom image, indicating that sufficient power was delivered to this area to excite the red fluorescent protein with which they were labelled. One of them looks nearly contiguous with the magenta cell since this is a maximum intensity projection. Again, we encourage the reviewer to look at the supplementary data if they so desire, since these data are more easily viewed in 3D.

But overall we agree, in many cases the ray optics model is not so much worse. However, it is pretty clear that it is variable in its quality, and this variability has clear experimental disadvantages (phototoxicity/missed fluorescent objects). It would occasionally perform very

badly on parts of certain samples (a tendency we did not see with neural network excitation over dozens of experiments). An example of this can be seen in the highly over-illuminated region on the bottom right of the image below.

Because of these shortcomings, we concluded that a more flexible model that incorporated optical physics and a neural network was a clearly better choice

On a different note, since our last revision we've created a detailed tutorial that uses entirely open source hardware and software to implement our technique (as the reviewer pointed out in the previous revision would be a major hurdle for practitioners). We feel the best way to put these questions to rest about the best models is for practitioners to try it themselves on their own model systems.

Reviewer #3 (Remarks to the Author):

In the revised manuscript and the response to review, several points are clarified. There are some remaining issues.

1, The authors claim that the machine learning model is much improved compared to the physical optics model (based on the surface profile and a constant decay coefficient). Although the lymph node has a heterogeneous structure, the scattering coefficient does not have any sharp variation within the typical imaging depth. As the authors claim the machine learning method is superior, I hope the authors can quantify how much the machine learning outperforms the simple physical optics model. Machine learning is a buzzword. In many important applications, it indeed outperforms other approaches. However, based on the nature of the application and the presented data, I am not convinced that it significantly outperforms physical optics models. Based on the acquired data, I hope the authors can quantify the improvement (is it 2x, 20x better or just 20% better?).

Putting the terminology of “physical optics” and “machine learning” aside for a moment, it is worth noting that, statistically speaking, both models are doing the exact same thing: learning

the parameters of a model from data, and then using those learned parameters to make predictions on new data. Both models are also designed based on relevant physics of the problem. So this isn't really a choice between machine learning and physics, but rather a choice between two models that utilize both learning and physical priors.

Every statistical model comes with its own inductive bias (the tendency to produce certain classes of solutions). We agree with the reviewer that a model based on first principles of physics of the problem is likely a better starting point, because it will have a better inductive bias for this problem. Our first attempt to solve this problem used such a model for this reason. However, it quickly became clear that this model was far too computationally intensive to be calculated while scanning through a sample in real time.

As a result, we needed to solve the challenge of coming up with a new model that was able to be executed in real time, and we tested two possibilities: 1) A *mismatched* physics-based model, that incorrectly assumes the sample is a sphere and calculates the physically correct solution for this different problem (but has the advantage that it can be pre-computed before the experiment because we don't need to know the sample shape). 2) A small neural network that approximates the true, physics-based solution. It's worth noting this neural network bears little resemblance to contemporary deep learning approaches, since it has 1,000-100,000x fewer free parameters. The behavior of these smaller networks is well-understood and it has been a standard statistical tool for nonlinear regression since the 1980s. Since it has access to the correct path lengths excitation rays travel through the sample to the focal point, it too is a physics-based model. It is simply approximating a computationally expensive numerical integration with much faster matrix multiplications for a >1000x speed improvement.

If one believes that a ray optics model correctly captures all of the relevant physics of the problem, then we know that the ray optics model we present data from comes to an incorrect answer, because it is solving the wrong problem (i.e. a differently-shaped sample). How bad its predictions are depends substantially on the shape of the sample being imaged and what part of that sample is being imaged. Our experience bears this out--the data shown in figure 2 shows visible heterogeneity of standard candle cell brightness compared to the neural network model. In other experiments using the mismatched ray-optics model, we would find subregions of the lymph node (usually near curved edges) where its predictions were way off (see saturated region in image below). We did not see such errors with the neural network model. Quantifying this effect on one or a small number of datasets is not especially informative, since its error changes depending how much each sample deviates from the assumption of being a perfect sphere, which can vary substantially in lymph nodes since that change sizes and shapes regularly under different biological conditions.

We also believe the reviewer has misinterpreted the strength of our claim of how much better the neural network model performs. Our claim (line 143) was only that “Given these limitations, we found that a physics-based neural network was a better solution to this problem.” We don’t believe neural networks outperform pure physics-based models in general (in fact, quite the opposite) or that there is a massive difference in performance in the present data. However, our data do support the conclusion that it is more robust and consistent than a mismatched pure-physics model in this case, and it is certainly much easier to work with computationally.

In addition to the data presented in figure 2, the data in the supplementary movies shows homogeneous signal that allows us to effectively track cells as they move throughout the volume. These data are representative of a large number of experiments showing the same, while the performance of the mismatched ray optics model was generally inconsistent, and occasionally gave extremely incorrect predictions like the one shown above. We have rewritten the description of the models in the text to make these issues more clear.

Ultimately, the best way to come to definitive answers to the questions about improvements is for competing models to be tested by researchers in multiple contexts. To facilitate this, we’ve created a detailed tutorial since our last revision that uses entirely open source hardware and software to implement our technique.

2, The authors claim that the presented method can compensate for the absorption of emission light (line 284). Is there any experimental data that can back up this claim? If so, please add this to SI. If not, please consider removing this claim.

Thank you for pointing this out. This claim was a little unclear and has now been revised. We don’t mean to say we know that emission light is being absorbed. Merely that the model makes no distinction between scattered excitation light and absorbed emission light (since both lead to an attenuation of signal). We don’t know for sure if absorption of emission plays a role, but the compensation (i.e. increasing incident power) is the same in either case. We only point this out because an absorption of emission light would imply depth-dependent photobleaching and phototoxicity, which is an important potential issue for practitioners of our method to be aware of.

Reviewers' Comments:

Reviewer #1:

Remarks to the Author:

My comments on the revised version of this manuscript focus on the two concerns highlighted by the editor; "whether their updated measurements of improved imaging volume are reasonable, and whether their data do in fact show the benefits over physical optics models given the curved shape of the lymph nodes."

1. The authors do indeed provide a detailed comparison of effective imaging volume of their approach (LAMI) versus other approaches that could be utilized with current 2P microscope technology (Supplemental note 1 and Fig S6a). I think this represents their argument well, and in retrospect feel that I did not give sufficient weight to these simulations in my comments on the previous version of the manuscript. There cannot be a single number that defines the improvement, as it is highly dependent on the morphology (curvature) of the specimen, and on which excitation strategy is being used for comparison. The important point is that for many specimens, such as small lymph nodes, LAMI will provide a very appreciable enhancement of imaging volume; and beyond merely an increased volume will enable visualization of other – likely morphologically and functionally distinct – regions that would otherwise be inaccessible.

2. The issue of whether, and by how much the neural network model improves on a physical ray optics model was raised by referees 2,3. I would only comment that I think this is a secondary issue, which should not factor into decision as to publish the paper. The revised manuscript now includes a more clear description and comparison of these models, and notes specific instances where a pre-computed ray optics model would fail.

Overall, the revised manuscript incorporates useful new clarifications and explanation. Moreover, the production of a tutorial video will assist the adoption of this methodology which should prove valuable to many immunoimaging researchers.

Reviewer #3:

None

REVIEWERS' COMMENTS

Reviewer #1 (Remarks to the Author):

My comments on the revised version of this manuscript focus on the two concerns highlighted by the editor; “whether their updated measurements of improved imaging volume are reasonable, and whether their data do in fact show the benefits over physical optics models given the curved shape of the lymph nodes.”

1. The authors do indeed provide a detailed comparison of effective imaging volume of their approach (LAMI) verses other approaches that could be utilized with current 2P microscope technology (Supplemental note 1 and Fig S6a). I think this represents their argument well, and in retrospect feel that I did not give sufficient weight to these simulations in my comments on the previous version of the manuscript. There cannot be a single number that defines the improvement, as it is highly dependent on the morphology (curvature) of the specimen, and on which excitation strategy is being used for comparison. The important point is that for many specimens, such as small lymph nodes, LAMI will provide a very appreciable enhancement of imaging volume; and beyond merely an increased volume will enable visualization of other – likely morphologically and functionally distinct – regions that would otherwise be inaccessible.

2. The issue of whether, and by how much the neural network model improves on a physical ray optics model was raised by referees 2,3. I would only comment that I think this is a secondary issue, which should not factor into decision as to publish the paper. The revised manuscript now includes a more clear description and comparison of these models, and notes specific instances where a pre-computed ray optics model would fail.

Overall, the revised manuscript incorporates useful new clarifications and explanation. Moreover, the production of a tutorial video will assist the adoption of this methodology which should prove valuable to many immunoimaging researchers.

Thank you. A tutorial for implementing LAMI can be found at:

<https://doi.org/10.5281/zenodo.4314107>